# Adaptive Protein Tokenization

**Rohit Dilip** [1]   **Ayush Varshney** [2]   **David Van Valen** [1]

## Abstract

Tokenization is a promising path to multi-modal models capable of jointly understanding protein sequences, structure, and function. Existing protein structure tokenizers create tokens by pooling information from local neighborhoods, an approach that limits their performance on generative and representation tasks. In this work, we present a method for global tokenization of protein structures in which successive tokens contribute increasing levels of detail to a global representation. This change resolves several issues with generative models based on local protein tokenization: it mitigates error accumulation, provides embeddings without sequence-reduction operations, and allows task-specific adaptation of a tokenized sequence's information content. We validate our method on reconstruction, generative, and representation tasks and demonstrate that it matches or outperforms existing models based on local protein structure tokenizers. We show how adaptive tokens enable inference criteria based on information content, which boosts designability. We validate representations generated from our tokenizer on CATH classification tasks and demonstrate that non-linear probing on our tokenized sequences outperforms equivalent probing on representations from other tokenizers. Finally, we demonstrate how our method supports zero-shot protein shrinking and affinity maturation.

## 1. Introduction

Advances in generative modeling have transformed our ability to understand the sequence-structure-function relationship for biological molecules. The advances made are particularly striking for proteins, as current methods can fuse amino-acid sequences with structural and functional

[1]California Institute of Technology [2]Carnegie Mellon University. Correspondence to: Rohit Dilip <rdilip@caltech.edu>.

*Proceedings of the 43rd International Conference on Machine Learning*, Seoul, South Korea. PMLR 306, 2026. Copyright 2026 by the author(s).

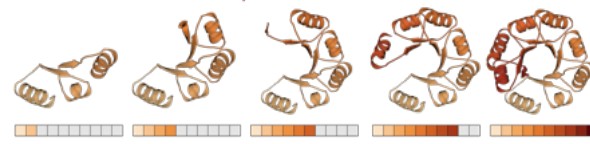

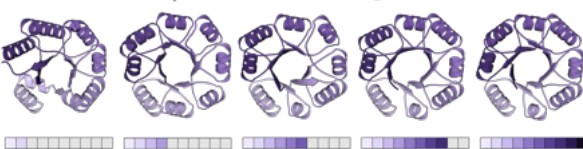

*Figure 1.* Prior work uses a single token to represent a local neighborhood around each residue. In our approach, every token provides additional global information, allowing large proteins to be compressed using fewer tokens

.

information to learn rich multi-modal representations of proteins (Wang et al., 2025a; Hayes et al., 2025). These methods are currently used for applications ranging from the generation of new therapeutics to de novo enzyme design (team et al., 2024; Ahern et al., 2025). One challenge these methods face is the diversity of biological data modalities. For instance, protein sequences are sequences of amino acid identities, protein structures are sequences of three-dimensional coordinates, and protein function is best described in natural language (Alberts et al., 1994).

Tokenization provides a tractable way to combine these modalities. A tokenizer maps each modality into a set of discrete tokens, which are then used for generative modeling (Van Den Oord et al., 2017). This approach is appealing because it maps the representation-learning problem into a space where one can draw on the successes of multi-modal models in other domains (Sun et al., 2024). The current paradigm for protein structure tokenization uses locality: tokenizers create a sequence of tokens by pooling information from spatially neighboring locations along a protein sequence (Hsieh et al., 2025). While these tokenizers have demonstrated the capacity for high-fidelity reconstruction, they have two problems. First, the generative models they enable considerably underperform their non-tokenized counterparts, in part due to error accumulation – a single missampled token can lead to large changes in a predicted

structure (Hsieh et al., 2025; Zhang et al., 2024a). Second, locality-based tokenization scales poorly. The number of tokens scales linearly with protein size, so building models that can operate over *many* proteins (e.g., protein complexes with large numbers of components) becomes computationally expensive (Hayes et al., 2025; Wang et al., 2024).

In this work, we pursue the alternative approach of global tokenization, where each token provides additional *global* context. This approach is inspired in part by computational methods such as the Fourier and wavelet transforms, which decompose a signal into coarse (low-frequency) and fine-detail (high-frequency) components. In computer vision, efforts in representation alignment have demonstrated that the strength of a generative model strongly depends on qualities of the underlying representation (Leng et al., 2025; Yu et al., 2024); similarly, it is natural to question whether global representations may improve generation quality. A global tokenizer would also encode large but simple proteins like tropomyosin using fewer tokens than their amino acid lengths would suggest, leading to better scaling.

Concretely, we develop a model (APT, or **A**daptive **P**rotein **T**okenizer) that dynamically tokenizes global structure. Each token acts in a coarse-to-fine hierarchy, providing additional information and refinement to a global representation rather than corresponding to a specific spatial neighborhood. We build our tokenizer using a scalable diffusion transformer architecture (Peebles & Xie, 2023; Rao et al., 2019; Orengo et al., 1997) without explicit symmetry biases and benchmark it on reconstruction and representation learning tasks. Using tokens from APT, we train an autoregressive model to generate proteins and study the quality of the generated proteins. Across all metrics, we match or outperform state-of-the-art models. Our primary contributions are as follows:

1. We introduce an adaptive diffusion-based tokenizer for biological structure in which tokens represent global descriptors rather than local neighborhoods.

2. We validate this model on generative, representation learning, and reconstruction tasks. In generative and representation learning tasks, we demonstrate significant improvements over existing models that rely on local structure tokenizers. On reconstruction tasks, we show that this tokenizer achieves performance comparable to SOTA models while enabling controllable compression.

3. We introduce inference-time techniques that use token entropy as a heuristic for sample complexity, enabling more principled sampling based on information loss and mitigating the effects of error exposure.

4. We demonstrate how our approach to conditioning enables zero-shot applications such as protein-shrinking and affinity maturation.

## 2. Background and prior work

**Tokenization:** A common paradigm to generate continuous data is to train a *tokenizer*, which maps continuous inputs to a finite, discrete codebook (Van Den Oord et al., 2017; Esser et al., 2021; Sun et al., 2024). Once continuous data has been compressed to discrete tokens, an autoregressive or discrete diffusion model can be trained just as in natural language. In the last two years, machine learning methods in biology have trained tokenizers to describe protein backbone structures, which enable the creation of multimodal sequence-structure models (Wang et al., 2024; Hayes et al., 2025; Gaujac et al., 2024; Gao et al., 2025) (see Appendix A for an accessible background to biology). Early works relied heavily on SE(3) invariant losses and architectures derived from seminal protein folding work in AlphaFold2 (Jumper et al., 2021). More recent works have embraced learning stochastic invariance/equivariance via data augmentation and scalable architectures (Abramson et al., 2024; Geffner et al., 2025; Dilip et al., 2025). Discrete approaches have consistently underperformed continuous diffusion models on generative metrics; however, a key difference is discrete approaches provide useful representations for downstream tasks and can integrate smoothly into multimodal models.

**Flow matching and diffusion autoencoders:** While tokenizers historically relied on a reconstruction loss, recent work has proposed replacing this with a diffusion or flow matching objective. In flow matching, a neural network learns to approximate a vector field corresponding to the flow, which maps two probability distributions under a pushforward operation. Integrating the learned vector field allows one to sample from distributions defined by data (Lipman et al., 2022; Liu et al., 2022). Both flow matching and diffusion have been widely used for biological generation (Bose et al., 2023; Yim et al., 2023; Watson et al., 2023; Abramson et al., 2024; Hayes et al., 2025). In a diffusion autoencoder, a model encodes data to a sequence of tokens which *condition* a generative diffusion process (Preechakul et al., 2022; Sargent et al., 2025; Chen et al., 2025), rather than directly reconstructing the data. While diffusion autoencoders pay a heavy inference cost (as one must now integrate an ODE or SDE to reconstruct data), they potentially obviate the need for semantic losses (see Appendix K for discussion on this point and on the difference between flow matching and diffusion).

**Adaptive tokenization:** A number of recent computer vision works have studied ways to develop tokenizers that are *adaptive* in the sense that the number of tokens produced can be tied to image complexity. CAT uses a caption-based system to model image complexity using caption

information (Shen et al., 2025). ALIT uses a recurrent system where 2D grid-tokens are recurrently resampled to a variable-length 1D sequence (Duggal et al., 2025). Most similar to our efforts is FlexTok, which uses nested dropout and a flow decoder to create adaptive tokens (Bachmann et al., 2025).

## 3. Method

**Architecture:** APT is a diffusion autoencoder with a discrete bottleneck (Figure 2). We take as input a length $L$ sequence of raw C$\alpha$ coordinates normalized to have zero-mass. A bidirectional attention transformer maps raw positions to a latent sequence $\mathbf{c} \in \mathbb{R}^{L \times d}$ (Vaswani et al., 2017). We discretize to $\hat{\mathbf{c}}$ using finite-scalar quantization (FSQ) with levels (8,5,5,5) (effective codebook size 1000) and use these tokens to condition a diffusion decoder trained using a flow-matching objective (Mentzer et al., 2023). To impose adaptivity, we uniformly sample an upper cutoff $U(1, \ldots, \min(L, k_{\max}))$ and apply nested dropout (Rippel et al., 2014). This setup encourages the model to place more critical global information in the first few tokens and more detailed information in later tokens. We use relative positional encodings throughout and share adaLN parameters across all decoder layers as done in (Dilip et al., 2025). The use of a diffusion decoder allows us to make use of stochastic equivariance where symmetries are learned, rather than enforcing symmetries at the encoder level.

To encode sequence length for generative purposes, we project the protein size from the first token. This is consistent with the adaptivity constraint, as protein size and other global properties are zero-frequency information in the sense they do not vary at all with sequence index. The protein size is trained with a cross entropy loss and the size is teacher-forced in the diffusion reconstruction. The full loss is then

$$\mathcal{L} = \mathcal{L}_{\text{flow}} + \lambda_{\text{size}} \mathcal{L}_{\text{size}} \qquad \mathcal{L}_{\text{flow}} = \|(\mathbf{x} - \epsilon) - \mathbf{v}_\theta(\mathbf{x}_t, t)\|^2 \tag{1}$$

with $\mathcal{L}_{\text{size}}$ a cross entropy loss regressed from the first token of $\mathbf{c}$. We find that setting $\lambda_{\text{size}} \approx 0.01$ provides very accurate sizes (generally within two amino acids) without detracting from the loss.

We have two stages of training. First, we train an autoencoder as described above augmented with random rotations to learn stochastic equivariance, similar to models like Proteina and AlphaFold3 (Geffner et al., 2025; Abramson et al., 2024; Wang et al., 2025b). Second, we train GPT-style autoregressive models over APT tokens to evaluate generative capabilities (Radford et al., 2018; 2019). We discuss training in Appendix C.

**Data:** We train on synthetic AlphaFold2 predictions from the Foldseek clustered AFDB database (van Kempen et al., 2022; Varadi et al., 2022). This clusters the full AFDB and keeps a single representative structure for each cluster. After filtering for quality and size (described in Appendix F), we are left with $\approx 473,000$ structures.

**Autoregressive generation:** We use standard nucleus or min-p sampling to generate a sequence of tokens that conditions the diffusion decoder (Nguyen et al., 2024). Due to the adaptivity constraint, for a sequence of generated tokens $\{\mathbf{c}_i\}_i^L$, any contiguous subsequence $\{\mathbf{c}_i\}_i^N, N < L$ can be passed to the model and will be a valid approximation to the underlying generation. This allows us to drop the tail and delegate the responsibility of finer details to the diffusion decoder, which addresses the observation that small levels of atomic noise introduced via error exposure are often sufficient to severely reduce designability. This approach trades representation fidelity for generative capacity, and the number of tokens to drop depends on the task at hand (e.g., representation learning tasks that strongly depend on small details may use more tokens). For the simple task of unconditional structure generation, we explore three distinct strategies. In *finite cutoff* sampling, we simply keep a fixed number of tokens for all samples. In *entropy cutoff* sampling, we sample tokens up to a fixed per-token entropy value. Finally, in *minimum entropy sampling*, we fit a spline to the entropy curve and take the cutoff corresponding to the first local minimum. Entropy based sampling methods have been explored in prior work; they are particularly valuable here because every prefix serves as a valid conditioning signal (Zhang et al., 2024b).

**Diffusion decoding:** Standard ODE-based sampling integrates the learned vector field, i.e., integrates

$$d\mathbf{x}_t = \mathbf{v}_\theta(\mathbf{x}_t, t) \, dt \qquad \mathbf{x}_0 \sim \mathcal{N}(0, 1) \tag{2}$$

It is common to instead integrate the following SDE, which has identical marginals at $\eta = \gamma = 1$.

$$d\mathbf{x}_t = \mathbf{v}_\theta(\mathbf{x}_t, t, \hat{\mathbf{c}}) \, dt + g(t)\eta \, \mathbf{s}_\theta(\mathbf{x}_t, t, \hat{\mathbf{c}}) \, dt + \sqrt{2g(t)\gamma} \, d\mathcal{W}_t \tag{3}$$

Standard practice is to bias this integration towards higher probability regions by reducing $\eta$ (score annealing), which trades diversity for designability.

**Classifier annealing:** Prior works have observed that while classifier-free guidance leads to better prompt adherence, shifting away from the true data manifold can lead to artifacts (Geffner et al., 2025). Because biomolecules are very sensitive to small amounts of noise (even noise that does not significantly affect RMSD), we sought to mitigate this issue by annealing the flow/score fields between

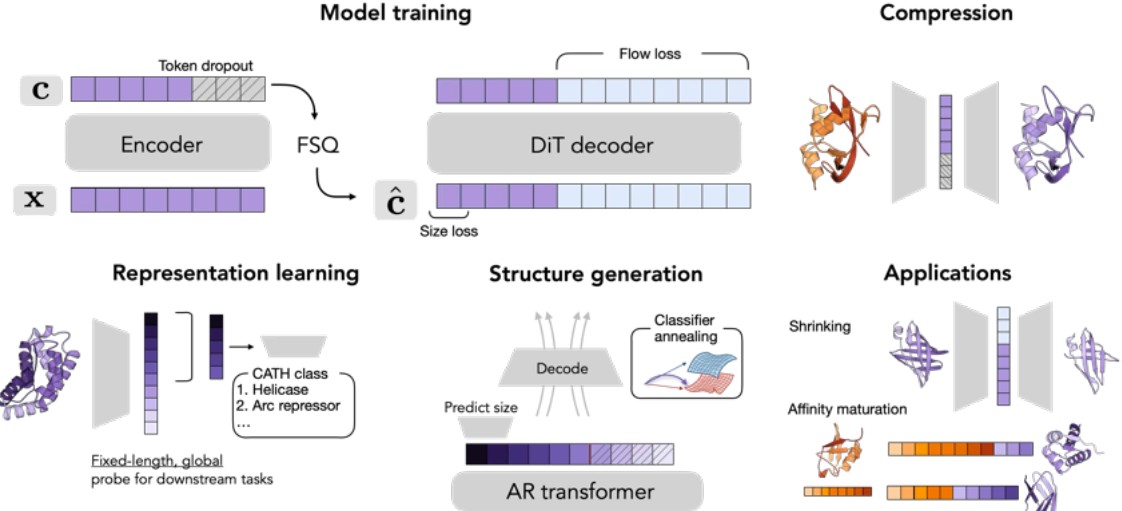

*Figure 2.* Overview of our approach. **Model training:** Raw input coordinates pass through a transformer encoder to create a 1D sequence of conditioning latents. These are discretized and pass condition a diffusion decoder. The noised coordinates are used in a flow loss objective, and the protein size is regressed from the first few latents. **Compression:** To compress a protein, we encode it and drop tokens from the tail up to a desired reconstruction. **Representation learning:** Our approach freely provides fixed size, global representations of proteins for downstream tasks, in contrast with most tokenizers that require a mean-pooling operation. **Structure generation:** During structure generation, we regress the protein size from the first 1-4 tokens, then use the size and conditioning tokens to decode the atomic coordinates. We optionally perform classifier annealing and drop out the tail based on entropy heuristics, providing a scaled method to balance representation fidelity and faithfulness to natural sequences. **Applications:** Decoupling protein size from conditioning leads to several immediate applications, such as protein miniaturization and affinity maturation.

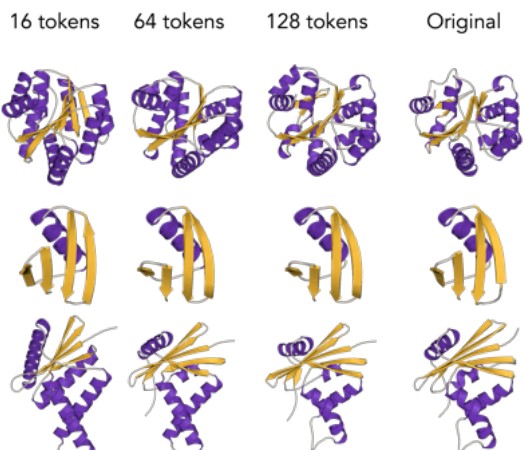

*Figure 3.* Reconstructions at varying conditioning strengths, colored by secondary structure content. The number of conditioning tokens increases from left to right and by hue. As the number of tokens increases, more disordered secondary structure emerges. For particularly simple details, 16-32 tokens often captures much of the required resolution. Additional visualizations in Appendix H.

conditional and unconditional guidance during decoding. This combines limited intervals and manifold constrained guidance (Kynkäänniemi et al., 2024; Chung et al., 2024), and generalizes a late-stage refinement step used in several works (Faltings et al., 2025; Raghu et al., 2025). We parameterize classifier annealing as follows, with $\alpha \in [0, 1]$

$$
\begin{aligned}
\mathbf{v}_\theta(\mathbf{x}_t, t) = {} & \mathbf{v}_\theta(\mathbf{x}_t, t \mid \varnothing) \\
& + (1 - t^\alpha)\Big(\mathbf{v}_\theta(\mathbf{x}_t, t \mid \hat{\mathbf{c}}) - \mathbf{v}_\theta(\mathbf{x}_t, t \mid \varnothing)\Big),
\end{aligned}
\tag{4}
$$

## 4. Evaluation

### 4.1. Reconstruction

We first evaluate APT on reconstruction using RMSD, TM-score, and rFID. RMSD and TMscore are standard; FID was introduced in (Geffner et al., 2025) (there termed FPSD) and applied to tokenization in (Dilip et al., 2025). We use three held-out test datasets: CAMEO, a subset of CATH, and a subset of the AFDB. We exclude structural homologs of all test sets from our training sets. In addition to DPLM2 and ESM3, two frontier multimodal models with structural tokenizers, we compare against Kanzi (a diffusion tokenizer), IST, and FoldToken (both GNN-based) (Dilip et al., 2025;

| Method | CATH | | CAMEO | | AFDB | |
|---|---|---|---|---|---|---|
| | RMSD | TM | RMSD | TM | RMSD | TM |
| DPLM | 1.64 | 0.897 | 1.65 | 0.876 | 4.68 | 0.810 |
| IST | 1.20 | 0.940 | 1.64 | 0.916 | 2.87 | 0.862 |
| Foldtoken | 1.30 | 0.920 | 2.54 | 0.881 | 2.16 | 0.858 |
| ESM3 | 1.05 | **0.957** | **0.86** | **0.955** | 2.38 | 0.915 |
| Kanzi | 1.09 | 0.937 | 0.98 | 0.941 | 1.18 | **0.936** |
| Ours | **0.90** | 0.941 | 0.90 | 0.941 | **1.17** | 0.929 |
| Ours (128,TF) | 1.94 | 0.880 | 1.79 | 0.884 | 1.94 | 0.880 |
| Ours (128,LS) | 1.94 | 0.880 | 1.82 | 0.903 | 1.96 | 0.906 |

*Table 1.* Reconstruction quality metrics on CATH, CAMEO, and AFDB with 512 samples each. Using the full tail, the differences between the best performer and ours are $< 0.1$ Åin RMSD and 0.01 in TMscore. Using 128 tokens still provides strong reconstructions by both RMSD and TMscore. Learning the size (LS) has minimal impact on RMSD, as in the worst case it is generally less than 4 residues off from the teacher forced (TF) case.

Gaujac et al., 2024; Gao et al., 2025). The latter two do not include generative capabilities.

An important assumption in our approach is that we *do not want to optimize for reconstruction*. Simply having good reconstruction does not necessarily make a useful downstream model (Zhang et al., 2024a; Hsieh et al., 2025). We characterize reconstruction performance as a necessary but insufficient condition for good downstream performance. Table 1 summarizes our model's performance on reconstruction. By taking the full tail, we are on par with or outperform many continuous diffusion models. Throwing away the tail slightly harms reconstruction but allows us to favorably tune the model for downstream tasks (Section 4.2).

When the full tail is included, our approach performs similarly to well-tuned local tokenizer approaches. As expected, keeping more tokens lowers the RMSD. In Figure 4, we show the reconstruction performance as a function of the number of tail tokens in Figure 4. We can achieve sufficiently strong RMSDs for good generation ($\sim 2 - 3$ Åas observed in prior works (Wang et al., 2024)) after just 32-64 tokens ($< 2$Å), which affirms the compressibility of these structures.

### 4.2. Generation

We show generative results in Table 4 and visualizations in Appendix D. We study the following inference-time interventions: the tail dropout algorithm, classifier annealing, and sampling method. When evaluating generation results, we need to be cautious since dropping out a large number of tokens or taking $\alpha \to 0$ simply leaves one with a powerful unconditional diffusion model, which artificially boosts designability. We study inference speed in Appendix B.

Table 2 shows that dropping the tail at 16 tokens to mitigate error exposure yields a designability of 0.48, which is on par with leading discrete diffusion models like DPLM2. Adding additional classifier annealing boosts designability performance to $> 80\%$. When we add classifier annealing with entropy sampling, we can attain designabilities around 0.871, as shown in Table 4. To benchmark against posterior collapse, we also report the fraction of classifier annealed samples with TMscore $> 0.5$ as compared to no-classifier annealing. This ensures that the diffusion decode retains the same fold, rather than hallucinating a completely different protein. These generative results are a product of both the diffusion decoder and the adaptivity; APT outperforms diffusion autoencoders like Kanzi on designability, which require best-of-N sampling techniques to reach 0.60 designability. The appropriate number of tokens is task dependent; for instance, cryoET has much lower resolutions than binder design. Our approach provides a semantic vocabulary to smoothly tune reconstruction fidelity.

During generation, any prefix is a valid sequence, so we can freely stop at any point. In line with our earlier discussions of intrinsic complexity, the appropriate place to halt generation depends on the complexity of the generated sample. This motivates stopping criteria based on per token entropies. Table 3 presents a subset of these results. Explicitly, for a length $L$ sequence of probability vectors $p_{ik}$ (vocabulary index $k \in \{1, ..., V\}$) the cutoff mark $K$ is given by

$$\text{Finite: } K = \operatorname{argmin} \left\{ m \middle| - \sum_{k}^{V} p_{mk} \log p_{mk} < H_{\text{cutoff}} \right\}$$
$$\text{Spline: } K = \operatorname{argmin} \left\{ \lceil t \rceil \middle| H'(t) = 0 \right\}$$

(5)

where $H(t)$ is a continuous spline fit to the full entropy rollout. The latter reduces token-to-token noise at the cost of a full token rollout, rather than halting during generation. While naively sampling based on entropy performs on par with simply dropping an equivalent number of tokens uniformly, combining entropy sampling with classifier annealing improves quality metrics (0.5 Å improvement in scRMSD and 0.10 improvement in designability). Intuitively, entropy samplers encourage the model to drop the all frequency modes higher than the ones reflecting the first point of uncertainty; classifier annealing is necessary to fill the dropped modes.

We also studied how token dropout affects distributional coverage. We find that taking more tokens reduces designability (more opportunities for error exposure), but improves gFID (distributional coverage). More details are in Appendix I. Adaptive tokens provides a controllable way to trade off exploration for exploitation. Lower token counts correspond to higher individual sample quality, while higher

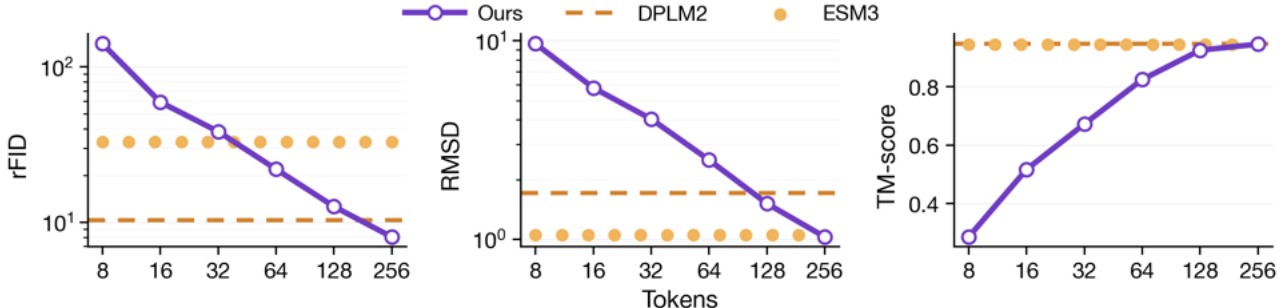

*Figure 4.* Impact of the number of tokens on reconstruction quality. We compare against DPLM-2 and ESM-3, the two other tokenized models that demonstrate generative capabilities. Across rFID (left), RMSD (center), and TMscore (right), all metrics improve with additional tokens. The x-axis is the maximum number of tokens used; e.g., for 32 tokens, APT uses at most 20 tokens to encode a protein of length 20.

*Table 2.* (Top) Impact of token count under standard inference (left) and classifier annealing (right, $\alpha = 0.8$). Dropping the tail improves performance by removing missampled tokens at the cost of lower structural diversity. Classifier annealing compensates and uniformly improves designability. (Bottom) Effect of classifier annealing for fixed token cutoff. Increasing classifier annealing strength ($\alpha \to 0$) leads to improved performance both less fidelity to the prompt. While $\alpha \geq 0.8$ leads to a large percentage of proteins with the same fold, $\alpha = 0.3$ leads to complete collapse. Based on these observations, we default to $\alpha = 1.0$ corresponding to the linear ramp $1 - t$.

| Tok | sRMSD ↓ | Des. ↑ | Tok | sRMSD ↓ | Des. ↑ |
|---|---|---|---|---|---|
| 16 | 3.09 | 0.482 | 16 | 1.80 | 0.771 |
| 32 | 3.88 | 0.379 | 32 | 2.12 | 0.719 |
| 64 | 4.68 | 0.314 | 64 | 2.12 | 0.703 |
| 128 | 5.64 | 0.232 | 128 | 2.26 | 0.670 |

| $\alpha$ | sRMSD ↓ | Des. ↑ | TM |
|---|---|---|---|
| $\infty$ | 8.48 | 0.156 | – |
| 1.0 | 4.74 | 0.462 | 0.93 |
| 0.8 | 4.07 | 0.508 | 0.86 |
| 0.3 | 3.00 | 0.669 | 0.15 |

*Table 3.* Combining inference strategies yields emergent gains in performance. We explore using a fixed number of tokens, sampling up to a a finite entropy, or sampling the first minimum of a spline. In the latter case, we rescale to control for the average number of tokens. We observe roughly equal performance across all three methods (top). Adding classifier annealing, however (bottom), increases the gap between entropy methods and fixed cutoff sampling to 0.5Å and 0.10 designability points.

| Type | Tok ($\mu \pm \sigma$) | RMSD | Des. |
|---|---|---|---|
| Fixed | $16.0 \pm 0.0$ | 3.09 | 0.482 |
| Finite | $17.0 \pm 8.9$ | 3.35 | 0.494 |
| Spline | $16.9 \pm 6.2$ | 3.04 | 0.533 |

| Type | Tok ($\mu \pm \sigma$) | RMSD | Des. |
|---|---|---|---|
| Fixed | $16.0 \pm 0.0$ | 1.80 | 0.771 |
| Finite | $17.0 \pm 8.9$ | 1.35 | 0.871 |
| Spline | $16.9 \pm 6.2$ | 1.31 | 0.867 |

token counts more thoroughly explore the full distribution of protein structures.

## 4.3. Representation learning

Prior studies into representation learning for tokens focus on local residue probing tasks (e.g., epitope prediction) (Yuan et al., 2025) or pairwise interactions (e.g., contact prediction) (Hayes et al., 2025). However, many realistic fine-tuning settings involve predicting global tasks using bespoke data from an in-house experiment (e.g., solubility, thermostability). We are interested in probing the representation strength of *global* representations which may track protein function better. We perform linear and MLP probing

on the CATH classification task (Rao et al., 2019). Results for topology-level folds are presented in Figure 5, with additional results in Appendix E, with further results in the Appendix. A nontrivial challenge in protein representation learning is going from variable-length representations ($L \times d$ for protein of length $L$) to a fixed length representation. Standard approaches generally mean-pool residue level representations (going from $L \times d \to d$), which can remove critical information, operates differently on proteins of different lengths, and tends to scale poorly on downstream tasks with protein size. For this reason, more sophisticated approaches based on optimal transport have been proposed (NaderiAlizadeh & Singh, 2025). With APT tokens, because every token provides additional global information, *any* prefix of arbitrary size is a valid representation. Thus, we can simply take a fixed number of tokens for every protein without any pooling. We directly use the discretized coordinates in FSQ; that is, for a model with a maximum of $N$ tokens

*Table 4.* Generative metrics. An autoregressive model trained on APT tokens outperforms other discrete sequence models and approaches the performance of state-of-the-art diffusion models. We compare against discrete diffusion models (DPLM2, ESM3), third-party autoregressive checkpoints trained using the structure tokenizers, and autoregressive models trained using local diffusion decoders (Kanzi). We also compare against best-of-N performance (k=4) to demonstrate the upper limit in autoregressive performance. See Appendix N for details.

| Method | Des. ↑ | scRMSD ↓ | Div. ↓ | Nov. ↓ |
|---|---|---|---|---|
| DPLM2 | 0.486 | 3.31 | **0.300** | **0.615** |
| ESM3 (32 steps) | 0.270 | 17.51 | 0.535 | 0.804 |
| ESM3 (256 steps) | 0.258 | 15.72 | 0.552 | 0.781 |
| DPLM (AR) | 0.320 | 8.99 | 0.297 | 0.656 |
| ESM (AR) | 0.520 | 4.25 | 0.275 | 0.615 |
| Kanzi | 0.562 | 3.78 | 0.517 | 0.779 |
| Kanzi (best-of-$N$) | 0.617 | 3.65 | 0.511 | 0.786 |
| Ours | **0.871** | **1.352** | 0.395 | 0.774 |

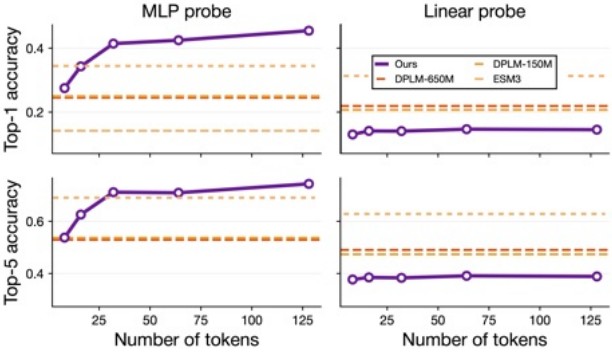

*Figure 5.* MLP (left) and linear probing (right) results on the CATH-T classification task. We plot top-1 (top) and top-5 (bottom) accuracy. An MLP probe outperforms probing using DPLM2 and ESM3. A linear probe results in weak performance. Our approach provides a convenient way to acquire fixed size representations without squashing information through a mean-pool.

and a quantization layer with $K$ levels, we map a protein $\mathbb{R}^{L \times 3} \to \mathbb{R}^{NK}$ *for all values of L.*

Figure 5 reports top-1 accuracy on CATH classification as a function of $K$. We compare to ESM3 and DPLM2. While linear probing underperforms models like ESM3 and DPLM2, our MLP probing results are stronger. Because our non-equivariant autoencoder entangles pose and structure, it stands to reason that they are not linearly separable, but the addition of a non-linearity is sufficient to extract useful global embeddings. It is noteworthy that for the CATH classification task, even highly compressed representations (down to 16 tokens) outperform larger models.

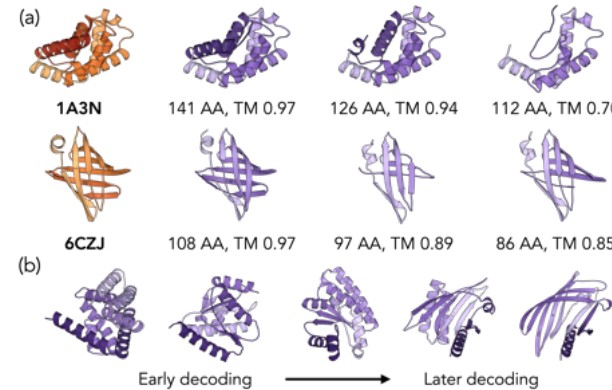

*Figure 6.* (a) Proof-of-concept shrinking experiments on a hemoglobin (1A3N) and beta barrel (6CZJ) protein. We track the number of amino acids and the TM score of the shrunken (purple) compared to the original (orange). These represent 100%, 90%, and 80% of the original residue size respectively (the TM drop at 100% comes from tokenization). (b) Inference-time scaling with beta sheet percentage as reward. Running longer beam search both expands the representational capacity (see Section 4.1) and searches over more solutions.

## 4.4. Applications

This section presents proof-of-principle experiments of some early applications enabled by adaptive tokenization. **Protein-shrinking:** The ability to shrink proteins while preserving functionality is a useful therapeutic capability (e.g., for better cellular penetration) (Devkota et al., 2024; Baron et al., 2025). A simple way to shrink a protein is to take the same conditioning sequence and condition a diffusion process containing a smaller number of residues. We emphasize that this strategy is plausible only because our tokenizer decouples the size of the protein from the size of the conditioning sequence. Figure 6a shows visual examples of source proteins and shrinking attempts, with the TM score to the original and amino acid count underneath each image. This approach preserves much of the global and local structure. An important caveat is much of protein function is mediated by sidechains and amino acid composition, neither of which our tokenizer accounts for.

**Test-time scaling:** A common task in structural biology is affinity maturation, where a starting sample with low affinity is known and we want to evolve the same towards high affinity. Given a protein with weak fitness (e.g., a weak binder), we encode it and use the prefix as a prefill to continue generation. Because any prefix provides a valid way to generate a protein, we can use inference-time scaling to search for candidates based on global rewards. We use our autoregressive model coupled with beam search, and maintain promising beams based on external reward function. We demonstrate this capability in three regimes. (1) We increase beta sheet content. (2) We generate a CATH

class using a verifier trained entirely on latent tokens. (3) We generate strong binders starting from a weak putative binder. We show beta sheet results in Figure 6b, and show the remaining experiments in Appendix L.2. A particularly appealing prospect is using verifiers trained on latent space tokens instead of performing full ODE integrations for each reward evaluation, which we demonstrate in Appendix L.2.

### 4.5. Experiments and ablations

In addition to the experiments performed earlier around generative design, we present several engineering ablations related to design choices. Plots corresponding to each ablation are in Figure 7. Further ablations are in Appendix I.

**RMSD does not always correlate with flow loss:** Surprisingly, we found that a lower flow loss does not guarantee a lower RMSD. While monitoring RMSD during training, we found a crossover point where flow loss continued to decrease but RMSD began to sharply increase. One possibility is our noise schedule (adopted from (Geffner et al., 2025)) is suboptimal for reconstruction.

**Scaling codebook size improves performance:** Increasing the codebook size from 1000 to 4000 unambiguously improves reconstruction across token cutoffs (Figure 7b). We also trained autoregressive models on codebooks trained with different maximum token cutoffs and did not observe significant differences (e.g., a generative model trained on a tokenizer with maximum 128 tokens performs similarly to a model trained on a maximum of 64 tokens) after accounting for token cutoffs. We therefore present the 128 token model as our base model, as it has better reconstruction performance without losing generative capabilities.

**Model size can bottleneck :** We also explored scaling the codebook size in the depth/width of the encoder. Our base APT-1k and APT-4k codebooks use two layer, 256 dimension encoders. We explore scaling the encoder dimension to 384 and the number of layers to four. As shown in Figure 7c, while all three 4k codebooks outperform the base 1k codebook, the weakest encoder actually has the strongest performance. There are a variety of possible confounders here. Our dataset is fairly small, and it seems reasonable that the encoder, decoder, and codebook all have to be scaled in tandem.

**Size loss weighting:** We find a very mild size loss weight ($\lambda_{size} \in [0.005, 0.01]$) is sufficient over the course of training. See Appendix L.2 for details. An important design detail is the use of absolute positional encodings on the input, rather than RoPE (see Appendix C.2).

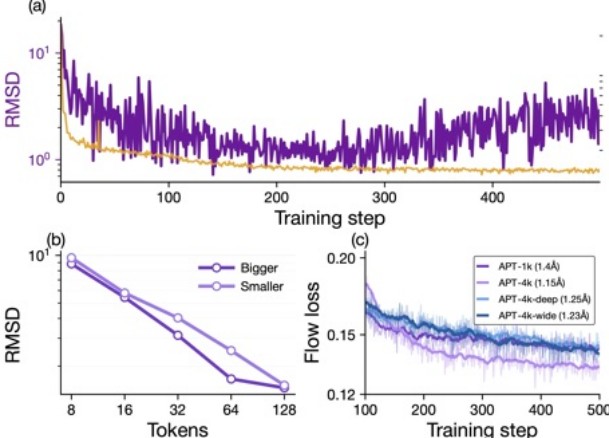

*Figure 7.* Ablations. (a) **RMSD does not always correlate with flow loss**. Although the flow loss (gold) continues decreasing, the reconstruction RMSD (purple) decreases after approximately 50,000 steps, then increases noticeably. (b) **Scaling codebook size improves reconstruction:** Swapping to a larger vocabulary containing 4375 tokens (over the length 1000 codebook used in most of our experiments) uniformly improves performance. (c) **Stronger encoders have ambiguous performance:** Although all three 4k codebooks outperform the base 1k codebook in RMSD reconstruction, the base 4k encoder outperforms both wider and deeper 4k encoders in *both* flow loss and RMSD reconstruction.

## 5. Outlook

In this work, we have presented an adaptive tokenizer and accompanying generative model for proteins. When applied with careful inference techniques, ours is the first autoregressive model that achieves quality metrics competitive with large scale diffusion models. Our key insight is to use nested dropout to enforce adaptive tokenization, where each additional token provides fine-grained *global* information. In addition to generation, this provides an effective way to compress proteins to a fixed dimension vector and enables several downstream applications.

Global tokenizers can provide a practical, task-aware approach to scale biological tasks; for instance, to model the dynamics of larger protein complexes on the order of $10^3$ residues. We have benchmarked against the structure generation capabilities of ESM3 and DPLM2, which are multimodal models trained on many sequences; using APT tokens in a multimodal setting is of substantial interest. Our work carries several limitations. Tokens from APT are useful for global tasks but ineffective for local tasks like motif scaffolding. Biology occurs at multiple length scales, and generative models must be capable of acting on both local and global considerations. Developing representations capable of reasoning across length scales and modalities will be a key challenge for frontier bioengineering.

## Impact Statement

This paper presents work whose goal is to advance generative models in biology. While novel protein based therapeutics hold the potential to benefit humanity, it is important to be aware that there are harmful applications of such generative models (e.g., around biosecurity). For this reason, the authors strongly support the principles outlined in Responsible AI x Biodesign.

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

## A. Overview of generative biology

*This section is a very brief overview of biology oriented towards machine learning practitioners.*

Proteins are structures that perform essentially all tasks required to sustain life, from immune responses to muscle contraction. They are formed by linked chains of amino acids. There are 20 amino acids of relevance. A remarkable fact about evolution is that almost all proteins of relevance to humans are formed from chains of these 20 amino acids, analogous to how a relatively small vocabulary of words can form an almost infinite number of coherent sentences.

This also implies that proteins can be represented by their sequence (i.e., a sequence of tokens from a vocabulary of length 20, with variable length depending on the protein) or by their structure (the 3D structure that the sequence "folds" into). AlphaFold2 addressed the protein folding problem; i.e., given a sequence, it outputs the structure of the protein. Models like ProteinMPNN (Dauparas et al., 2022) address the reverse problem – given a structure, sample plausible sequences.

Proteins are of interest for several reasons. From a machine learning perspective, their sequential structure makes them highly amenable to machine learning tools, many of which are oriented around sequence to sequence tasks. From a biological perspective, proteins represent possibly the largest source of validated data. The success of (Jumper et al., 2021) and the generative models that followed have inspired a great deal of belief in the possibility of proteins as a medium for intelligent therapeutic design.

In terms of data, every amino acid has a different set of atoms. However, all amino acids have the same set of backbone atoms – the nitrogen, carbon-alpha, and carbon atoms. For this reason, proteins are often represented using a tensor with shape $(L, 3, 3)$, corresponding to (sequence index, backbone index, position) respectively. A common workflow is to train structure generative models (like APT-AR, Proteina, FrameFlow, RFDiffusion, etc.), then use an inverse-folding model like ProteinMPNN to get a sequence that approximately folds into the desired structure. More recently, several models (Lin et al., 2024; Geffner et al., 2025) have focused on $C\alpha$ only generation, the same strategy we take in this work. Standard practice uses ProteinMPNN in $C\alpha$ only mode to perform inverse folding.

Despite the vast diversity of the natural universe, proteins adopt essentially three types of "secondary structure," which refers to the local substructures proteins form. There are highly structured alpha helices, flat beta sheets, and finally more disordered coils. See Figure 8 for examples of proteins with the secondary structures colored in indigo, gold, and gray respectively. Protein sequences are much more common than structures. Of the 200 million known sequences, only a few hundred thousand have a gold standard structure. Data distillation via folding models like AlphaFold2 provide an effective way to scale up structural data. Because the alpha helices are by far the most structured, AlphaFold2 often outputs alpha helices, so distilled structure models can have biases towards overproducing alpha helices.

Validating novel machine learning methods is challenging, since one cannot simply look at a protein and determine whether or not it is real/good. The standard methods (described in Appendix G) involve taking a putative structure, inverse folding it, refolding it, then using a self-consistency metric. For quite some time, the goal of generative protein models was to sample proteins that even "looked" plausible, akin to image models from 2016 to 2019 (Salimans et al., 2016). More recently, various works are considering how to shift towards generating proteins that accomplish a specific function.

## B. Inference

One advantage of autoregressive models is they are very fast compared to diffusion models, which require many passes of full bidirectional attention to integrate a full ODE/SDE. In contrast, autoregressive models can use kv-caching to accelerate sampling, but they lack bidirectional attention which is believed to be quite important for proteins.

We find that our approach admits the strengths of both approaches. The coarse-to-fine hierarchy of APT tokens means autoregressive ordering is quite natural. When generating conditioning tokens, we (1) can use kv-caching and (2) can generate a short sequence, using finite entropy to determine stopping. Surprisingly, we find that we can use only 20-40 diffusion timesteps during decoding, a decrease of 10x from Proteina. We attribute this partially to the observation that the conditioning tokens make early timesteps much more informative, but defer a more careful investigation to future work.

# C. Training

## C.1. Training procedure

As discussed in the main text, we inject the structural tokens by concatenating them with the noised sequence, then backpropagating only through the coordinate indices in the concatenated sequence. In code, this reads:

```
x_t = x_1 * t + (1 - t) * x_0
z_t = Linear(x_t)

z_ctx = encoder(x_1)
z_ctx = quantize(z_ctx)

latents = cat([
    z_t,
    cfg_mask(nested_dropout(z_ctx)),
])

out = decoder(latents)
v = out[:len(z_t)]
```

The operation 'nested_dropout' stochastically masks the sequence of latents. We also drop the entire conditioning sequence with probability 0.1 to enable classifier free guidance.

In the second stage of training, the autoregressive language model is trained on full token sequences. During inference, we optionally drop tokens using the heuristics described in the main text. We observed minimal difference between training on truncated vs untruncated sequences.

## C.2. Hyperparameters

All architecture and training hyperparameters required to reproduce our work are shown in Table 5. These include our autoregressive generative model (a,b) and our tokenizers (c,d). These settings represent our base model used for our generative results; we describe deviations from these settings in Section 4.5 and Appendix I. For the most part, these models do not seem particularly fragile to hyperparameter changes and the specific settings our final models used are somewhat arbitrary. We emphasize the following three design choices that are actually important during training. **Tokenizer training should be in float32** . Ambient diffusion does not work well in low precision. Surprisingly, we found that we needed float32 training even for the autoregressive model. **Models need sequence packing**, or they have very noisy gradients and struggle to converge. **The encoder and decoder need to scale commensurately**. Scaling one without scaling the other doesn't seem to help (e.g., we did not see benefits from pushing the encoder to 512 channel dimension). **Absolute positional encodings in the encoder.** We use RoPE (Su et al., 2024) in the decoder and generative transformer, but found that providing absolute embeddings during the autoencoder training was quite helpful for convergence speed. Finally, note that APT is pronounced Ah-Puh-Tuh.

# D. Generation

Visual samples from an autoregressive model trained on APT tokens are shown in Figure 8. All samples are designable. A notable failure mode is overprediction of alpha helices, which is a known phenomenon with models trained on synthetic data. Search protocols (as described here) or CATH conditioning (as described in (Geffner et al., 2025), which we leave to future work for the AR case) can be used to resolve this.

# E. Representation learning

This section describes our representation learning ablations. For language models like DPLM2 and ESM3, we mean pool the per-residue embeddings to go from a tensor with shape (batch, residue, channels) to (batch, channels), excluding special tokens (e.g., BOS, EOS, padding tokens) from the pooling operation. This is a standard approach, and is discussed at further length in (NaderiAlizadeh & Singh, 2025). For APT tokens, we take the first $K$ tokens, padding with zeros if the natural sequence length $L$ satisfies $L < K$. Every input gets encoded to a sequence of indices $p_{il}$, where $i \in \{0, 1, \ldots, K-1\}$ and $l \in \{0, \ldots, M-1\}$, where $M$ is the number of levels in FSQ (for us, 4 in most experiments) . We flatten this vector to attain a single input $\mathbf{h} \in \mathbb{R}^{KM}$. This is used as a fixed input to the model. Linear probing uses a single layer to the CATH vocabulary size, and MLP probing uses a Linear-Swish-Linear sequence.

A concern while MLP probing is the equivalency of parameters and FLOPs. While one can take various strategies to ensure

| (a) AR Architecture | | (b) AR Training | | (c) Tokenizer Architecture | | (d) Tokenizer Training | |
|---|---|---|---|---|---|---|---|
| Layers | 20 | Batch size[*] | 64 | Encoder channels | 256 | Batch size | 192 |
| Channels | 1024 | Learning rate | $1 \times 10^{-4}$ | Encoder layers | 2 | Learning rate | $3 \times 10^{-4}$ |
| MLP dilation | 4 | Adam $\beta_1$ | 0.9 | Decoder channels | 512 | Min LR | $10^{-4}$ |
| Attention heads | 16 | Adam $\beta_2$ | 0.95 | Decoder layers | 12 | Adam $\beta_1$ | 0.9 |
| Positional encoding | RoPE | Grad clip | 1.0 | FSQ Levels | 8, 5, 5, 5 | Adam $\beta_2$ | 0.999 |
| Max tokens | 128 | Min LR | $1 \times 10^{-5}$ | | | Grad clip | 10 |
| *252,893,184 parameters* | | Decay iters | 100,000 | *47,394,567 parameters* | | Micro steps | 2 |
| | | Micro steps | 4 | | | $\lambda_{\text{size}}$ | 0.01 |

*Table 5.* Model, training, and tokenizer hyperparameters. The effective batch size[*] in AR training is stochastic, since we use sequence packing to address variable-length sequences in proteins. Similarly, the batch size in training our tokenizer are all the same structure; every element of the batch is a different rotational view. We use gradient accumulation to provide

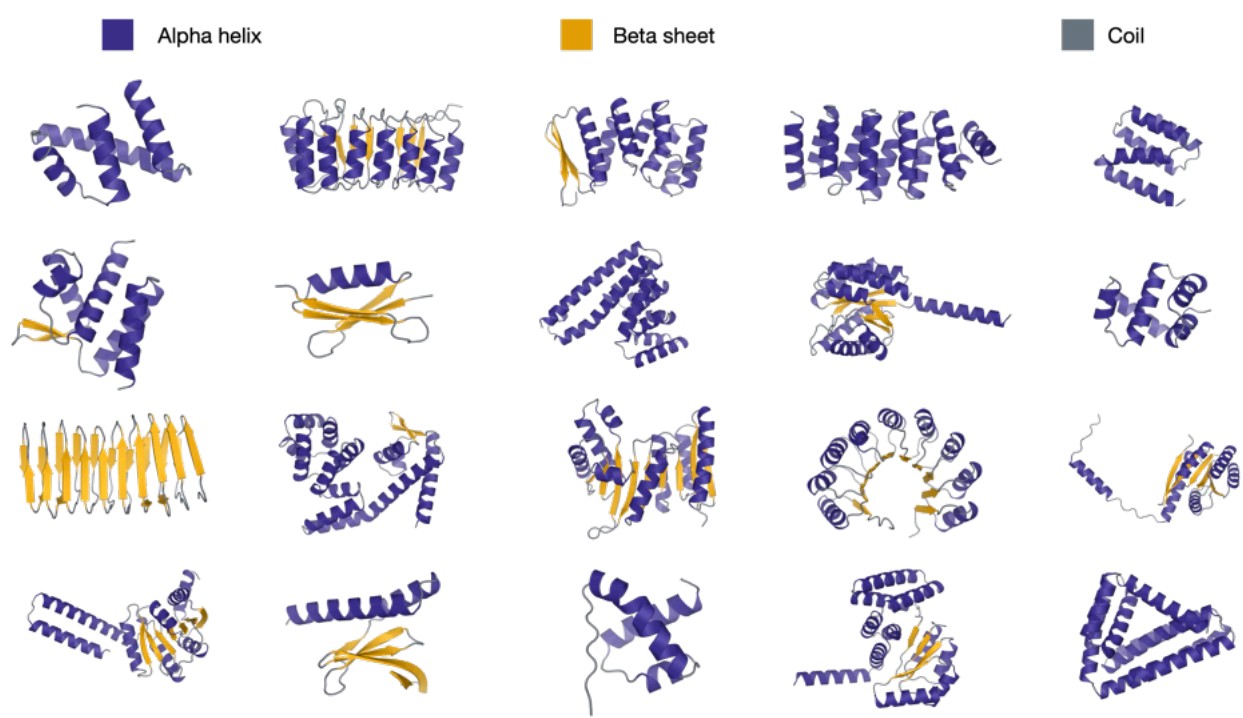

*Figure 8.* Samples from APT-AR colored by secondary structure. These visualizations are from finite-entropy based sampling with score annealing ($\eta = 0.3$, max entropy$= 2.0$). We find that unconditional sampling in this setting replicates known folds (e.g., repeated betas, partial TIM barrels). All samples are designable.

fair comparisons (e.g., projecting to a common basis), at 128 maximum tokens and 5 layers, APT representations have dimensionality of 640, which is significantly less than either ESM3 or DPLM2.

Figure 9 displays training curves on CATH classifiers across C, A, and T levels. Our global approach consistently outperforms larger tokenized models. We anticipate that on local probing tasks, our approach would underperform these models (where the input was a nearby token to the region of interest). As we pointed out in the main text, a large class of fine-tuning objectives are related to global properties. APT tokens are an appropriate solution for such tasks. We show additional representation learning tasks in Table 6, including ablations using attention pooling as a potentially more fine-grained method to compress information in ESM3/DPLM2.

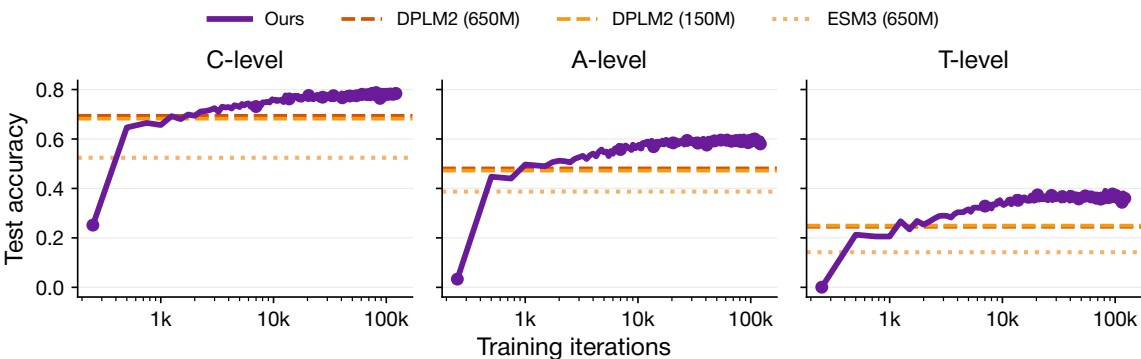

*Figure 9.* Test set accuracy on C, A, and T level training. Global tokens (purple) consistently outperform local tokens with mean-pooling.

| (a) Clustering results | | | |
| --- | --- | --- | --- |
| Method | Kept clusters ($\geq 8$) | Weighted TM mean | Frac. scored pairs TM > 0.50 |
| Foldseek | 217 | 0.514 | 0.506 |
| APT ($k = 32$) | 32 | 0.486 | 0.390 |
| APT ($k = 64$) | 64 | 0.500 | 0.415 |
| APT ($k = 128$) | 128 | 0.513 | 0.454 |
| APT ($k = 256$) | 256 | 0.540 | 0.518 |

| (b) Representation tasks | | | | |
| --- | --- | --- | --- | --- |
| Model | EC (Fmax) | Metal (Acc) | GO (Fmax) | CATH (Acc) |
| APT (MLP) | 0.310 | 0.52 | 0.29 | 0.46 |
| ESM (MLP) | 0.085 | 0.56 | 0.28 | 0.34 |
| ESM (Attention) | 0.073 | 0.56 | 0.28 | 0.36 |
| DPLM (MLP) | 0.130 | 0.62 | 0.26 | 0.25 |
| DPLM (Attention) | 0.080 | 0.57 | 0.27 | 0.26 |

*Table 6.* Numerical results for clustering and downstream representation tasks.

## F. Data

Our data pipeline follows (Dilip et al., 2025). We use the AFDB filtered using Foldseek clustering, and keep only proteins of size 256 or less. This is for computational reasons. During training, we mean center all data, work in nanometers, and randomly rotate each protein. We keep proteins with average plDDT > 80 and standard deviation plDDT < 15. We manually move homologs of proteins in any downstream test sets (CAMEO, CASP, a subset of CATH) to the test set.

## G. Metrics

This section describes our specific implementations of various metrics.

### G.1. Designability

We follow standard practices for designability. We use ProteinMPNN (Dauparas et al., 2022) in $C\alpha$ only mode to inverse fold 8 sequences with sampling temperature 0.1. We use ESMFold to fold each sequence and compute the self-consistent root-mean-square deviation (scRMSD) between the original structure and the output structure. Designability is the fraction of samples with < 2Å scRMSD. We report both the designability and the scRMSD score. Designability is particularly problematic since only 60% of natural proteins are actually designable, but the scRMSD of natural proteins is about 0.90 Å, so reporting both provides a more holistic quality metric.

### G.2. Diversity

We report the average inter-chain TMscore using TMalign (Zhang & Skolnick, 2005). As length normalization can unfairly bias the TMscore, we only take pairs within 16 residues of each other in size. This average is taken only over designable samples.

### G.3. Novelty

We use Foldseek to search against the PDB dataset and return the maximum TM alignment. We report the mean of all maximum alignments across designable samples.

## H. Additional reconstruction visualizations

Figure 10 shows additional reconstruction samples. These are curated for secondary structure diversity, but not by quality. Beneath each token count we show the TMscore to the original structure. For many folds, we can capture a TMscore greater than 0.5 using just 32 tokens. At 64 tokens, 99.4% of proteins have TMscore greater than 0.5.

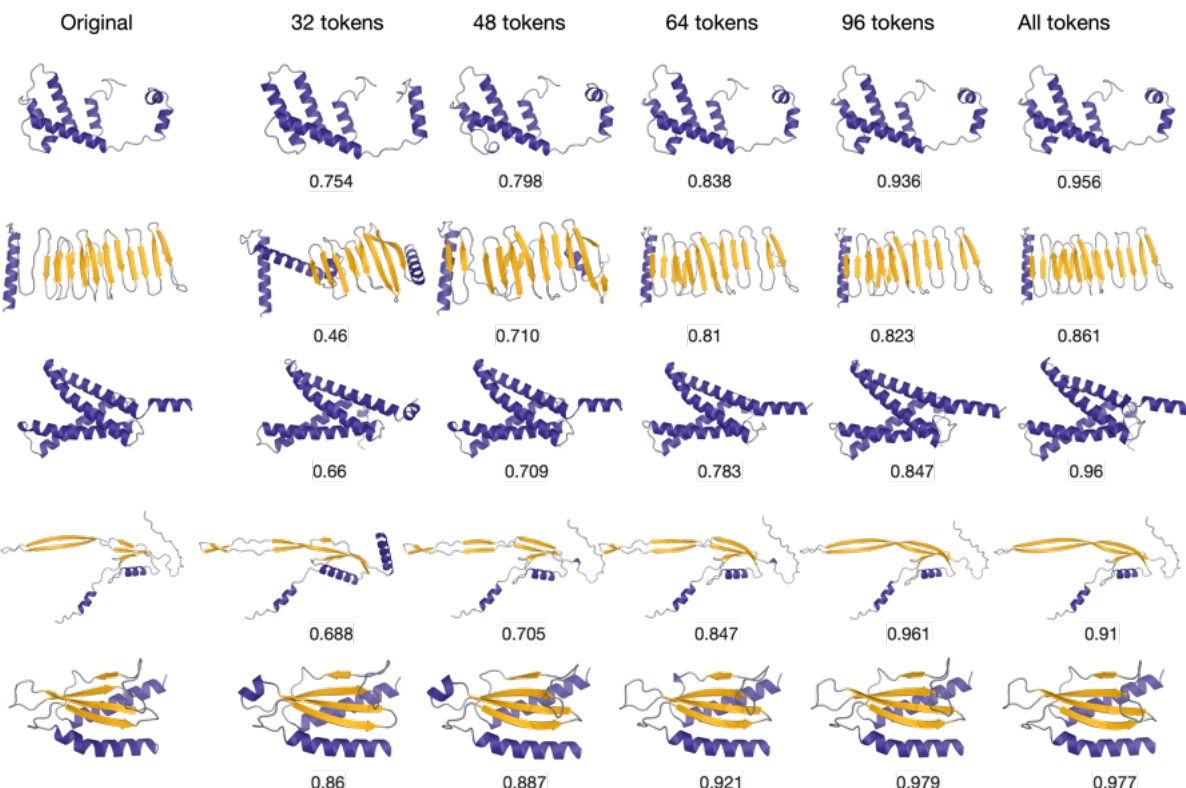

*Figure 10.* Reconstructions with ground truths on the far left and increasing number of conditioning tokens from left to right. We list TMscores against the original below each reconstruction.

## I. Ablations and additional experiments

This section contains additional ablations not described in the main text.

**Tokens lead to scalable generative models.** We train three separate models using the same codebook and plot loss and per-token accuracy in Figure 12. These models are 12 layers, 512 channels (small, 38M parameters), 16 layers, 768 channels (medium, 114M parameters), and 20 layers, 1024 channels (base, 250M parameters). We observe clear improvements for the same codebook (size 1k) by scaling the model. These results are in Figure 12. While these scaling results are promising, we emphasize that optimization metrics (e.g., loss, per token accuracy) are not sufficient to conclude better downstream performance; model performance may not improve monotonically with scaling (Handina & Mazumdar, 2024).

**Token effect on gFID.** We considered the impact of the number of retained tokens on gFID. The results are in Figure 11. As the number of tokens increases, gFID monotonically decreases. This is consistent with our observation that more tokens provides greater ability to represent complex secondary structures. Our results in Section 4.5 suggest that fewer tokens is

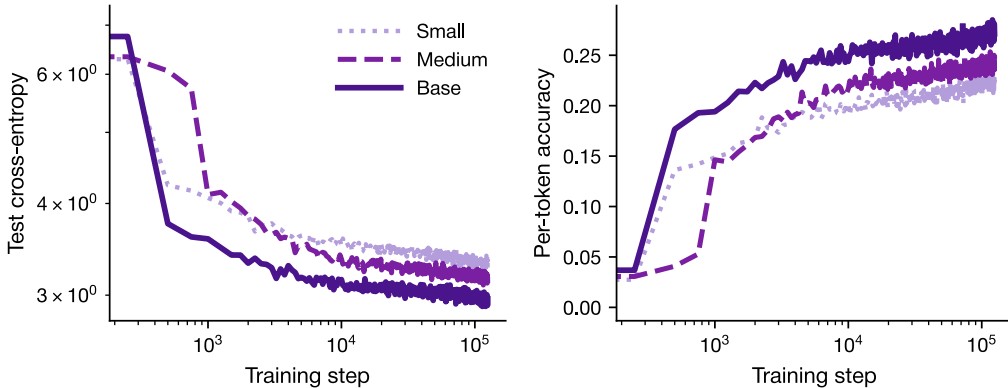

*Figure 12.* Scaling plots for language model with size 30M, 110M, and 250M (small, medium, base). (Left) Cross entropy loss on test set during training. Plot is log-log. (Right) Top-1 per token accuracy vs training steps

more useful for quality metrics like designability. This makes the tradeoff in protein complexity explicit. It also suggests (as has been acknowledged in numerous other places (see e.g., (Lu, 2025; Geffner et al., 2025)) that we need metrics that make this tradeoff in distribution coverage more explicit.

**Size losses.** Our final design uses a cross-entropy loss to predict the protein size. We ablate the strength of the size loss weight and show training curves in Figure 13. We find that $\lambda_{\text{size}} = 0.005 - 0.01$ is achieves excellent size loss without harming flow optimization.

**Effect of integration steps on RMSD.** Surprisingly, we find that we need relatively few inference steps to converge an integration. 50-75 steps suffices to achieve most of the final accuracy, and even $\approx 50$ steps achieves 90% of the final RMSD. This is shown in Figure 11.

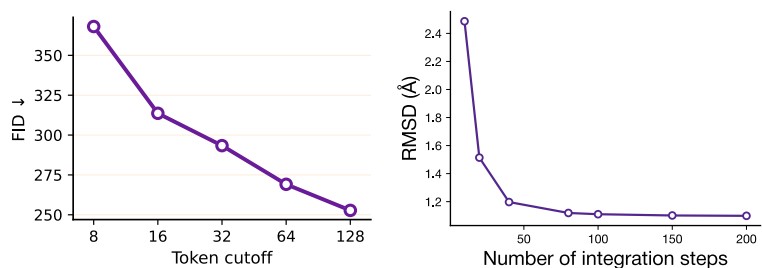

*Figure 11.* (Left) gFID vs number of retained tokens. (Right) Reconstruction performance vs number of integration steps.

**Maximum token cutoff.** We explored training tokenizers *and* downstream generative models with varying maximum token cutoffs at $k_{\text{max}} \in \{32, 64, 128\}$. We observed that there was not a huge difference in performance between the 32 token model and the 128 token model with only the first 32 tokens being used, in either generative capabilities or in tokenizer performance. We thus default to just using the 128 token model, since in the event that higher reconstructions are desirable it maintains that capability.

## I.1. Negative results and minutiae

Inspired by (Zhang et al., 2025) and (Redmon & Farhadi, 2018), we include a brief section on smaller observations that do not merit detailed ablations, but that may still be useful to the broader community.

**Register tokens for conditioning.** Register tokens are a much more satisfying way of conditioning, but still result in weak gradients. We tried using a wider and deeper encoder and they still died very quickly. We think this is probably solvable.

**Recursive size readout.** We tried to implement a readout of the size where one could get finer and finer predictions by sampling more tokens. There are some variants of this (e.g., each token binary searches towards the true size), but just reading out the size from the first token worked as well without really impacting performance.

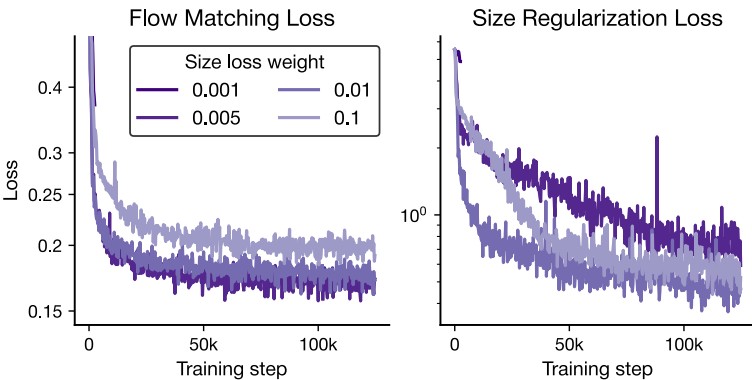

*Figure 13.* Flow/size loss for various values of $\lambda_{\text{size}}$. Setting $\lambda_{\text{size}} = 0.1$ harms flow loss significantly, but the remaining parameter settings all eventually perform equivalently. ALthough $\lambda_{\text{size}}$ has a large apparent gap in size loss, this generally does not significantly impact performance (e.g., predicting a size of 102 instead of 100).

**GPT training stability.**    While debugging something else, we actually did a bunch of ablations where we varied learning rate warmups, more gradient accumulation, and a larger gradient clipping threshold (10 instead of 1). We saw no difference between runs.

**Detached size loss.**    We tried to estimate the size loss without propagating gradients back to the main encoder trunk. This does not work at all.

**Cross attention.**    Cross attention seems to work really well for text-to-image models, but really wasn't helpful at all in our case and significantly underperformed in-context conditioning.

**Global models avoid repeated tokens**    An issue with models like Kanzi is the tendency for some models to repeat strings of tokens during encoding due to the use of sliding window attention (e.g., `[100, 2, 72, 2, 72, 2, 72, 99, 30,...]`. This is hard to quantify but easy to identify, and empirically seems to harm generation. The stochastic dropout seems to avoid this issue entirely.

## J. Classifier annealing

Part of the original motivation behind diffusion autoencoders was to obviate the need for perceptual and adversarial losses. Whether this is true is still a little unclear; (Sargent et al., 2025) uses perceptual losses during the training process but (Chen et al., 2025) does not.

Perceptual losses enforce semantic consistency, while adversarial losses create realistic textures, edges, etc. Without adversarial losses, an image of a desk and a chair could be accurately reconstructed but may contain artifacts that immediately render it unrealistic to human eyes. An equivalent effect can occur with biomolecules. Relatively small displacements in C$\alpha$ position can render molecules unrealistic but keep RMSDs low. One reasonable approach to avoid this might be to try to follow in computer vision's footsteps by training discriminators and fold classifiers, then incorporating those as part of loss functions. As (Chen et al., 2025) observes, this is highly task dependent and makes training challenging; for this reason, most modern works in computer vision simply defer to the parameters used in (Esser et al., 2021). Classifier-free guidance trades off prompt-fidelity for staying on-manifold. In the case of biomolecules, we care more about staying on manifold than we do about prompt fidelity (the perfect beta barrel is pointless if it's biophysically implausible). Classifier annealing is designed to address this challenge. We know that diffusion models are capable of placing atoms in biophysically plausible locations (Geffner et al., 2025). A missampled token during an autoregressive rollout would lead to incorrect placement, so instead we use the conditioning to do global atom placement and anneal towards the general manifold for finer details.

## K. Comments on diffusion and flow matching

In this section, we discuss two points – the difference between diffusion and flow matching and why they are a compelling choice for autoencoders. Historically, diffusion models and score-based SDEs preceded the development of flow matching as used in this work. For the particular case of coupling to a Gaussian prior, one can show that flow matching and diffusion are equivalent up to a reweighting of the noise schedule (Gao et al., 2024). This is why, despite being trained on a flow, we immediately can perform score annealing. For $\mathbb{R}^n$, the score is related to the flow via

$$\mathbf{s}(\mathbf{x}_t, t) = \frac{t\,\mathbf{v}(\mathbf{x}_t, t) - \mathbf{x}_t}{1 - t} \tag{6}$$

For this reason, we and others often refer to models like APT as diffusion autoencoders trained using a flow matching objective.

In practice, many diffusion models learn the noise (and the score is equivalent to the noise up to reparameterization). v-prediction in diffusion is similar to flow matching in that both v-prediction and the flow represent linear combinations over the added noise and the clean sample. This is important because it avoids instabilities at the end of sampling. While this is an issue of efficiency for images, it is extremely important for biomolecules. Because diffusion takes place in ambient space, there isn't an easy heuristic to clip the maximum absolute value during the diffusion process as was done in early diffusion works. This observation, the general ease of implementing flow matching, and the success of prior flow matching works (Geffner et al., 2025) all contributed to our decision to use a flow matching objective. The ablations in Section 4.5, however, suggest that for the case of an autoencoder, further gains could be realized by adjusting the noise schedule.

## L. Applications

This section discusses the applications we presented in the main text.

### L.1. Shrinking

Our shrinking experiments merit two additional comments. First, because we explicitly read out size from the first token, technically conditioning the diffusion process using the same tokens will immediately be out of distribution. In practice this ends up not mattering very much.

Second, we ultimately care about preserving function. Structure is a heuristic for this. Other shrinking methods are mostly based on sequence, and rely on evolutionary patterns to guess at which amino acids can be deleted without harming function. These paradigms have respective pros and cons. For functions that are particularly structurally interpretable (e.g., a beta barrel encasing a functional unit), working directly in structural space makes more sense. For functions where the exact mechanism is unknown, using sequence methods to determine which residues are more amenable to being removed may be a better strategy. However, it also seems unlikely that sequence edits that dramatically change the structure will also preserve function, since the functional mechanism would need to change as well. We think that joint models that can attend to both sequence and structure will be a fruitful avenue moving forward, and that the technique we propose where the conditioning is decoupled from the sampling process will be a potent tool in this direction.

### L.2. Inference time scaling

We explored three experiments to study inference-time scaling. In all experiments, we have a reward function $R(x)$ defined by the task at hand.

**Overview of beam search.** Let $p_\theta(x_{1:T}) = \prod_{t=1}^T p_\theta(x_t \mid x_{1:t-1})$ be an autoregressive model over sequences $x_{1:T} \in \mathcal{V}^T$, and let $R(x_{1:T}) \in \mathbb{R}$ be a task-specific reward. Beam search approximately maximizes the combined objective

$$\max_{x_{1:T}} \log p_\theta(x_{1:T}) + \lambda R(x_{1:T}),$$

by maintaining at each step $t$ a beam $\mathcal{B}_t$ of the top $B$ prefixes under a prefix score. Each prefix $x_{1:t}$ is expanded with all tokens $v \in \mathcal{V}$ and scored as

$$S(x_{1:t}) = S(x_{1:t-1}) + \log p_\theta(v \mid x_{1:t-1}) + \lambda \widehat{r}_t(x_{1:t}),$$

where $\widehat{r}_t$ is an optional shaped reward satisfying $\sum_t \widehat{r}_t(x_{1:t}) = R(x_{1:T})$ when the reward is decomposable. The beam is pruned to the top $B$ prefixes at each step, and the highest-scoring completed sequence is returned.

In our case, because every token provides global information, we simplify this slightly by setting

$$S(x_{1:t}) = \log p_\theta(v \mid x_{1:t-1}) + \lambda\, R(x_{1:t}),$$

where $R(x_{1:t})$ is a single reward function that acts on all tokens.

**Search is hard with local tokenizers**  Global tokenizers make search challenging for three main reasons.

1. Prefixes are out-of-distribution. Most tokenizers are trained on whole protein chains. A prefix is **not** a valid input to the tokenizer. For rewards that operate over ground truth structures, this makes search quite challenging, because the decoder is out of distribution during the entire rollout until the end state is reached.

2. Partial states are mismatched with most rewards. Even with perfect prefix decoding, the generated protein would only be a partial structure. Most reward functions we are interested in (e.g., affinity measurements, binding, etc.) operate over entire proteins, so the reward calculation would be unreliable.

3. Rewards are not continuous with search. A successful search protocol requires that the existence of a high reward state $R(x_{1:T})$ suggests high reward states also exist for $x_{1:K}, K > T$. Similarly, it is desirable that a low reward state $R(x_{1:T})$ implies states $x_{1:K}, K > T$ are also low reward. This needs to hold in some weak sense, since otherwise we don't have a good way to prune the tree. Because most rewards we care about are global, this is absolutely not true for local tokenizer rollouts. Even something as simple as "more beta sheets" can have 0 reward for the first third of generation, then suddenly increase later during search (e.g., 3QVO).

These issues all stem from the fact that most protein rewards are global, not local (or to the extent that they are local, they can occur quite deep into search). Our approach thus enables much more effective beam search, which we validate in three experiments.

**Beta sheet percentagè**  A common issue with generative models, particularly those trained using distillation, is the overprediction of alpha helices. Various works have explored ways to increase beta sheet percentage (e.g., Proteina conditions on CATH class). While CATH conditioning is an option, we defined a reward function based on the beta sheet percentage to search for high beta sheet percentage structures. We implement this using the `annotate_sse` function from `biotite`. At each step of beam search, we do a mini-rollout (20 steps) and keep only the beams with the highest beta sheet percentage. Structures are shown in the main text (Figure 6).

**CATH class**  One problem with test-time scaling in diffusion modeling is that most rewards must be computed on clean samples, which means one needs to do mini rollouts at every step. This can get expensive quickly. A reasonable alternative is to train models that compute reward functions directly on raw tokens. We showed in Section 4.3 that the encodings from our approach could be used to perform CATH classification. We trained MLP reward functions that operate over tokens, then use them to perform beam search during generation. In this setting, *we do not need to perform mini rollouts*. The downside is we need to train an additional model, but this is a simple MLP and trains to $> 80\%$ accuracy in less than 24 GPU-hours. Depending on the task and data availability, this could be easier or harder than creating a reward function in ambient space.

We train the MLP using stochastic masking, similar to our encoder. This is described in Algorithm 1. We show search results in Figure 14 for mostly $\alpha$, mostly $\beta$, and mixed generations. This search is quite fast, since the reward function does not require any diffusion decoding (the samples presented in the figure took a few minutes with a very wide beam search), so the bottleneck shifts to autoregressive rollout which is easy to optimize.

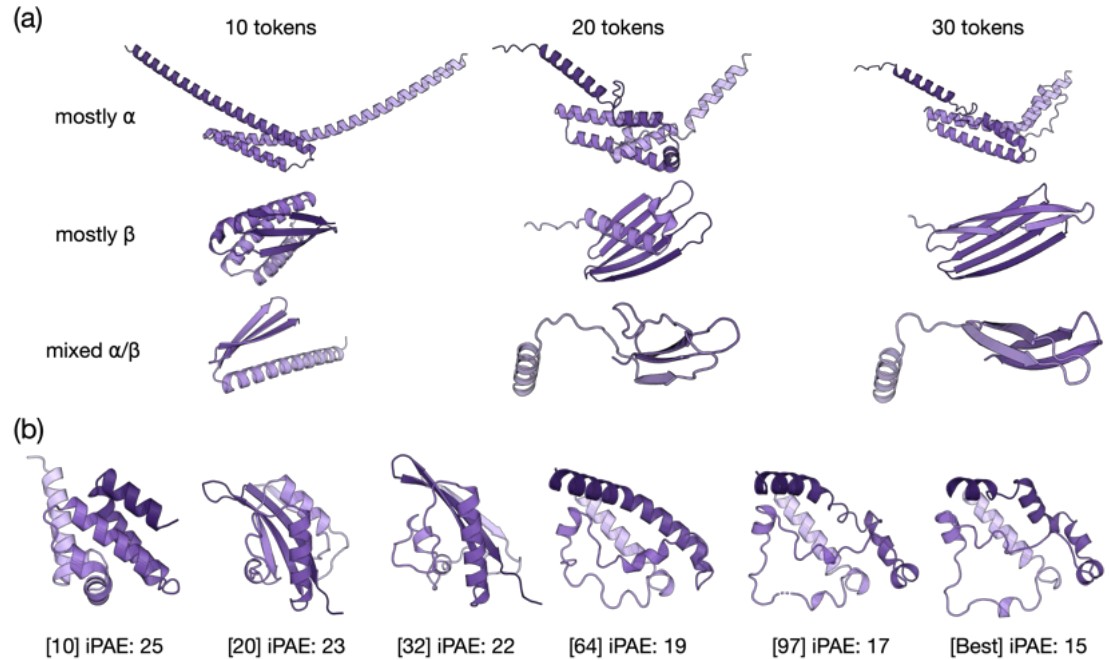

*Figure 14.* (a) Beam search using CATH reward classes in latent space. Each row is a different CATH class, each column is a different point along beam search. The token count has two implications. First, more tokens allows for more expressive secondary structure. Second, more tokens represents more search compute (i.e., a later point along search). The reward function is the probability of the desired class under our classifier. (b) Beam search using iPAE-based rewards. The token count is in square brackets beneath each structure. At later token counts, the majority of beams are based on the same high reward structure (i.e., they share a prefix), so so later structures look similar.

---

**Algorithm 1** Prefix-Masked Hierarchical Token Classification

---

**Require:** Protein coordinates $X \in \mathbb{R}^{L \times 3}$
**Require:** Target CATH label $y_{\text{tgt}}$
**Require:** Encoder $E$, classifier $f$
**Ensure:** Predicted label $\hat{y}$
 1: $(Z, \ell) \leftarrow E(X)$ {$Z \in \mathbb{R}^{M \times K}$ hierarchical tokens; $\ell \leq M$ valid positions}
 2: $m \sim \text{Uniform}\{1, \dots, M\}$ {Random prefix length}
 3: $\tilde{Z}_{i,:} \leftarrow \begin{cases} Z_{i,:} & \text{if } i \leq m \\ 0 & \text{if } i > m \end{cases} \quad \forall i \in \{1, \dots, M\}$ {Prefix masking via zeroing}
 4: $v \leftarrow \text{vec}(\tilde{Z})$ {$v \in \mathbb{R}^{MK}$ flattened representation}
 5: $\hat{y} \leftarrow f(v)$ {MLP classification head}
 6:
 7: **return** $\hat{y}$

---

**Algorithm 2** Affinity Reward via MPNN and AlphaFold-Multimer

---

**Require:** Target coordinates $X \in \mathbb{R}^{L \times 3}$, binder sequence $S$
**Ensure:** Reward $r$
 1: $\hat{S} \leftarrow \text{MPNN}(X; \tau = 10^{-6})$
 2: $p \leftarrow \text{iPAE}(\text{AFM}(S, \hat{S}))$
 3:
 4: **return** $r \leftarrow -p$

**Binder design** Our final experiment concerns binder design. In this setting, our reward function is defined by taking a putative structure, inverse folding it using MPNN, then using iPAE measured using AlphaFold-Multimer and cofolding the sequence to bind against. We use the `colabfold` implementation of AF-MM (Mirdita et al., 2022). We show examples in Figure 14b, where we start from a weak binder against PD-L1 and gradually mature it (decreasing iPAE in the process). This requires diffusion rollouts, for which we use 20 steps. The bottleneck is mostly in AlphaFold-Multimer.

We emphasize that iPAE is not a perfect reward. A number of works have explored binder design (Pacesa et al., 2024) and it is an open problem. Our intent here is to show that once an appropriate reward function has been designed (e.g., some combination of iPAE, solubility, etc.), our use of global tokens enables search methods for that reward. True de novo binder design requires sidechain awareness; our method (which more closely resembles pipelines like Bindcraft (Pacesa et al., 2024)) must start from a putative binder and improve affinity.

## M. Comments on tokenizers

**Reconstruction as a metric.** There is empirical evidence pointing out that having good reconstruction does not mean a model get better at the downstream tasks. For example, the ESM3 tokenizer does exceptionally well on CAMEO but does not seem to generate many designable proteins (possibly because it was trained on a lot of synthetic data, as suggested in (Geffner et al., 2025)). In computer vision, optimizing a reconstruction loss purely for a per-pixel MSE will lead to poor image generations. This motivates perceptual losses, which de-emphasize pixel exact accuracy and instead preserve semantic structure and textures. Similarly, adversarial/GAN losses encourage reconstructions that still look like real reconstructions. There are no pretrained networks that we can use to fill the role of these losses, and nobody has convincingly demonstrated that adversarial training is beneficial for biological design. Part of the appeal of diffusion tokenizers is they may remove the need for these losses. Our introduction of classifier-free annealing is an inference-time attempt to mimic the strength of adversarial losses. The main point, however, is that even in computer vision, we routinely sacrifice pixel losses for semantic clarity. Reconstruction is just a heuristic; the real optimization target is semantic clarity. We want proteins that fold consistently with well defined functions.

### M.1. Is APT really a discrete model?

We characterize our work as a discrete model. However, we have a diffusion process for decoding, so one can reasonably ask whether this is really a fair comparison. The main point, however, is that we get useful representations from our encoding. This is the entire reason to prefer discrete models; we can train larger, multimodal models and do information compression in a way that is not really feasible with diffusion models.

In our case, we clearly get good global representations. Diffusion techniques and classifier-free annealing certainly contribute to our strong performance. The sensitivity of protein design to atomic level details partially accounts for why diffusion is a useful technique. Our approach blends the strengths of both approaches – discrete representations for downstream tasks coupled with strong, atomically sensitive decoding. We note that the diffusion decoder cannot completely account for the strength of our model; we outperform Kanzi on generative benchmarks, which similarly uses stochastic equivariance and a diffusion decoder.

## N. Generation using other models

We generated 5,000 samples each from DPLM2 (150M and 650M) and ESM3-open for comparison. We computed the sequence length distribution from the CATH S20 non-redundant dataset, filtering to sequences $\leq 256$ residues. The distribution was binned into 7 bins: [0,64], [64,96], [96,128), [128,160), [160,192), [192,224), and [224,256]. For each bin, we used the midpoint as the representative length for generation. The number of sequences generated per bin was proportional to the bin's probability in the CATH distribution, ensuring our generated dataset matches the natural length distribution.

For DPLM2, we generate structures with temperature annealing from 2.0 to 0.1, stochastic unmasking with probability 1.0, temperature 1.0, and max iterations 500. For ESM3, we report structures using `GenerationConfig(track="structure")` with default parameters (temperature 1.0, cosine schedule, random strategy, temperature annealing enabled, top_p=1.0). We do not observe significant improvements in designability by increasing the number of steps. We report both 32 steps and 256 steps; the latter unmasks a single token per step. We observed quite low designability scores; this is consistent with benchmarks elsewhere (Geffner et al., 2025). For Kanzi,

we use the publicly available autoregressive checkpoint with default sampling parameters – minp with threshold 0.1 and $\eta = 0.2$. We use the DPLM/ESM-AR checkpoints provided in the Kanzi repo.

