# OpenReview forum: "Adaptive Protein Tokenization"
_ICML.cc/2026/Conference — ICML 2026 regular_

### Official Review · Reviewer_yymG · 2026-02-17

**Soundness:** 2
**Presentation:** 1
**Significance:** 3
**Originality:** 3
**Overall Recommendation:** 3
**Confidence:** 3

**Summary:**

This work proposes a new tokenization scheme for protein structure, with a central claim being that this tokenization scheme learns a coarse-to-fine-grained representation of protein structure (as opposed to each token capturing a local neighborhood). The authors evaluate the tokens on a series of downstream tasks, including reconstruction, generation, and representation learning.

**Compliance With Llm Reviewing Policy:**

Affirmed.

**Final Justification:**

Overall, the authors answered my questions constructively, and improved my doubts about the manuscript. I still find the presentation of the manuscript challenging, but I have improved my score from a 2 to 3.

**Key Questions For Authors:**

1. Please better explain how the architecture/methods results in learning of the desired representational properties, or better contextualize as a hypothesis that downstream experimentation tests.
2. Please better contextualize if the experiments are intended to test for properties, or serve as benchmarks, and if the latter, in particular clarify whether the generation benchmark is unfair to other methods given models are fundamentally given different information as prompt to generate from.
3. Please provide better baselines for the protein shrinking experiment and validate that the shrunken proteins are functional.
4. Please compare the method's ability to shrink representations for downstream representational purposes to other methods that do this.

**Limitations:**

Yes

**Strengths And Weaknesses:**

In general, the biggest weakness of this paper is its presentation (and I open with this to caveat that there may be important details I am missing in my review as a consequence). The paper is written assuming significant knowledge in both protein structure and generative models, meaning it is addressed towards a very targeted community - as someone who has adjacent knowledge on both (i.e. I am a bioinformatician who works largely with sequence-based understanding of proteins, and has some experience with generative models but now flow-matching specifically), many details are impenetrable to me. I sympathize with the authors because I can see how this arises from space constraints, but the paper could deeply benefit from more background motivating the model and experiments, and more accessible writing.

From this angle, I have concerns about some aspects of soundness - and I'm not sure if this is because the authors are truly making assumptions, or if these have been left unexplained in the current work. My questions are:
- I am not clear on how the architecture and training method results in "critical global information" being concentrated in the first few tokens, and "higher frequency information" being concentrated in later tokens. From what I understand, most of their loss is based upon the ability to reconstruct coordinates given a (variably) truncated latent sequence. Is the claim that this loss will enforce the desired distribution of information through tokens from first principle purely? If so, I can think of counter-arguments from principle also: suppose that there is enough information in the first few residues of a protein to reconstruct the rest of the structure (which might occur in proteins where the N-terminal is highly indicative of overall function, like a signaling peptide). Is there a stronger theoretical justification for why this loss induces this property? Alternatively - is this only a hypothesis that is validated by their downstream experiments? I can buy this too especially as many of their experiments do suggest this property emerges, but currently neither the writing of the methods nor the experiments frame as such.
- Related to the first point then - I am having a hard time interpreting if the downstream experiments are benchmarking results, or validation of the model's ability to learn coarse-to-fine grained token representations. This in particular confounds the interpretation of the generation benchmarks for me. I am leaning towards an understanding that the authors are claiming the former especially as the discussion emphasizes the quality of the model. My understanding is that these benchmarks are using the model in a partially-prompted way - given 16 starting tokens, generate a sequence. Are the other models also given equivalent information? But even if they are - given the authors' claim that their model learns a coarse-to-fine grained token representation whereas other models learn/use a local sequential representation - it seems that their model is being asked to conduct a fundamentally different task here - i.e. refining a coarse-grained structure into a fine-grained one. Could the authors clarify?

I also have a few comments about soundness independent as emerging from writing issues:
- For the protein shrinking application - the qualitative results alone are not convincing to me, and some of the structures (e.g. 1A3N) look to me close to what you'd expect if you just truncated the N or C terminal. To strengthen these results, what would the TM score and structure look like if they had actually truncated an equivalent number of residues as a baseline? Is there any way to assess if these truncated structures actually retain their function (e.g. through a downstream oracle)?
- How does the reduction of length result compare to other proposed methods (e.g. CHEAP, or dimensionality reduction-based techniques like DCTdomain)?

Overall, however, I am optimistic about the significance and originality of this work: if the authors improve their framing and contextualization of the work, I do think learning a protein structure embedding with the properties that the tokens represent coarse-to-fine grained structural information is of strong significance to the field.

---

> ### Author Rebuttal · Authors · 2026-03-30
>
> Thank you for your detailed reading of our work! We sincerely appreciate the clear effort you put into your review. While we strived to provide a level of background in line with norms in the field (see [1,2]), we recognize the challenges with evaluating cross-disciplinary works. We believe that most of your concerns can be resolved with some clarifications.
>
> **Q1: Adaptivity**
> The distribution of information is *not* enforced by some natural pre-existing structure; it is infused via nested dropout. Nested dropout ensures that the decoder always has access to the first token, almost never has access to the last token, and has intermediate access to the tokens in the middle. This encourages the encoder to put the most critical information with respect to our loss (a diffusion loss) in the first token and the least critical information (e.g., small perturbations that do not really affect the diffusion loss) in the last token.
>
> We can also give a theoretical justification. Suppose each token has capacity $K$, and for a sample $S$ with total information $K_S \gg K$, the encoder must distribute information across $L$ tokens. Let token $i$ contain information $K_i$. Under nested dropout, token $i$ is visible to the decoder with probability $p_i$, where $p_0 > p_1 > \dots > p_{L-1}$. The expected information available to the decoder is therefore $\mathbb{E}\left[\sum_i K_i\right] = \sum_i p_i K_i$. Since earlier tokens are retained with strictly higher probability, maximizing expected transmitted information under a per-token capacity constraint is equivalent to allocating information greedily to the earliest tokens first: fill token 0 up to capacity, then token 1, and so on. Thus, nested dropout induces an ordering where earlier tokens preferentially capture the most essential information, while later tokens encode progressively less critical residual details.
>
> The coarse-to-fine hierarchy is a _data-agnostic_ property from our training procedure. The example you give partially motivates this work; if the intrinsic information content of a protein is small, our work allows for fewer tokens to be allotted to the encoding (e.g., tropomyosin). This contrasts with local tokenizers where the tokenizer encoding is tied to the length of the protein.
>
> **Q2: Clarifications on generation**
> Reconstruction experiments demonstrate that the tokenizer learns a useful latent space that exhibits the desired "adaptive" structure. Generative experiments demonstrate that the tokens have useful internal structure for downstream tasks. This is standard in both the tokenization and generative modeling literature (e.g., rFID and gFID in computer vision).
>
> There is no "prompt" to the model aside from the initial \<beginning of sequence\> token. We instead generate an average of 16 tokens starting from scratch. With other models, a protein of length 200 requires 200 generations. With our approach, you can produce a length 200 protein by only generating 16 tokens. This is a product of the coarse-to-fine design, it is not a capability other models have. The models all have the same task (generate plausible proteins). The difference is because our approach requires fewer passes, there are fewer opportunities to make sampling errors. Our emphasis on generation quality is meant to affirm the value of the tokenizer.
>
> **Q3: Clarification on protein shrinking**
> A structure-only tokenizer can't preserve protein function; the purpose of these experiments is to argue that current tokenizers couple protein sequence length to latent encoding length, and this need not be the case. For example, one could repeat our dropout procedure but in an ESM embedding space to potentially acquire more functional proteins. We have clarified this in the main text; we're happy to move this to the appendix as we do not consider them to be a critical part of our work. Nonetheless, we think it is interesting and noteworthy to point out that the latent representation of a protein need not be tied to the sequence length.
>
> **Q4: Other methods:** CHEAP is an autoencoder that operates in the ESM2 *sequence* latent space. Our work is a structure tokenizer and encodes coordinates; the two models encode different data. CHEAP does not decouple the protein length from the encoding length (it downsamples then upsamples). Similarly, DCTdomain (1) applies on sequences, not structures, and (2) still flattens down the sequence axis; similarly to ESM2 and DPLM2, it flattens information and does not natively provide a global view of a protein.
>
> We once again thank you for your detailed comments. We have used your commentary to clarify and rewrite sections of the paper, and we believe that these points should now be more clearly expressed in the paper. We are happy to address any further concerns you may have.
>
> [1] Geffner et al, "Proteina: Scaling flow-based protein structure generative models"
>
> [2] Lipman, et al. "Flow matching for generative modeling."

---

> > ### Author Rebuttal · Reviewer_yymG · 2026-03-31
> >
> > Thank you to the authors for their comprehensive comments. In general, although their comments are convincing as to the validity of their experiments, I still struggle with their premise - my challenge with their response to my point about adaptivity is less about how information would be distributed in their learned token space, and more about how much information there is present globally about the entire protein structure given a limited window of coordinates. Their response operates on the assumption that the input gives total information - but my prior intuition here is that if you give, as input, only some coordinates of a protein structure, the remainder of the protein structure is not strongly encoded/predictable, particularly in extreme cases where only tens of coordinates are given. Hence - I am more concerned about the information even present in the input space, and thus the learnability of their representation, particularly in respect to generalization. Could the authors clarify?

---

> > > ### Author Response · Authors · 2026-04-03
> > >
> > > Thanks for the clarification. We want to highlight that the encoder **sees all the coordinates, not just a small section.** The encoder takes a sequence of protein coordinates and does **full self attention** across this sequence. The nested dropout at the end of the encoder encourages the encoder to move information to the first few tokens, but nonetheless, there is never a point where the model only seems a small section of the protein structure. The dropout occurs after several layers of all-to-all attention. Explicitly (with some simplifications), the code would look like
> > > ```
> > > Input: x_L3 # (sequence_index, coordinate)
> > > h_LD = Linear(x_L3) # (sequence_index, hidden_dimension)
> > > h_LD = TransformerEncoder(h_LD) # all to all self attention across all sequence indices
> > > h_LD = quantize(h_LD)
> > > h_LD = nested_dropout(h_LD)
> > > # ...flow matching loss + decoding
> > > ```
> > >
> > > What this means, as shown on the third line above, is the encoder always sees the entirety of the protein, never a small portion. Thus, there is no dependency whatsoever on how predictable a protein structure is; rather, what matters is the intrinsic structural complexity of the protein as a whole. The nested dropout intervenes after the model has had a chance to see the entire protein and build a representation over the entire protein.
> > >
> > > We hope these clarifications address the main concerns. If you find that the issues have been satisfactorily resolved, we would be grateful if you would consider updating your score accordingly.

---

### Official Review · Reviewer_L6AA · 2026-03-07

**Soundness:** 3
**Presentation:** 3
**Significance:** 3
**Originality:** 3
**Overall Recommendation:** 4
**Confidence:** 3

**Summary:**

This paper proposes Adaptive Protein Tokenization, a new way to represent protein structures for generative modeling. Instead of assigning tokens to local structural patches as in previous protein tokenizer, this method learns global, hierarchical tokens that progressively refine the entire protein structure from coarse to fine. The tokenizer is trained using a diffusion autoencoder with nested dropout, which encourages early tokens to encode global structure and later tokens to add fine geometric details. Experiments show that APT demonstrates strong structure reconstruction ability and significantly improves protein generation quality, achieving higher designability than prior structure tokenization approaches.

**Compliance With Llm Reviewing Policy:**

Affirmed.

**Final Justification:**

I am convinced by the author rebuttal reply and have no more questions. Overall speaking, I am now satisfied with the technical soundness and experimental verfication. I will keep my recommendation unchanged.

**Key Questions For Authors:**

1. How much does adaptive tokenization itself contribute to the performance gain compared to the diffusion decoder?

**Limitations:**

Yes

**Strengths And Weaknesses:**

Strengths
1. This paper introduces a novel perspective on protein tokenization that aligns token structure with global protein geometry.
2. The idea of leveraging nested dropout to enforce coarse-to-fine encoding is simple but powerful.
3. Experiment shows the tokenizer performs competitively or better across reconstruction, generative and representation tasks.

Weaknesses
1. It is still unclear whether the gains come primarily from the tokenizer itself or the diffusion decoding pipeline.
2. The scalability of the proposed model is limited with respect to model size.

---

> ### Author Rebuttal · Authors · 2026-03-30
>
> We are glad the reviewer appreciated the strength of our experimental results and the novelty of our approach. We’d like to address the primary weaknesses the reviewer raised.
>
> **Q1: Do the gains come from the diffusion decoder or the tokenizer?**
> We’re grateful to the reviewer for raising this point. This is a very valid question. It's a challenge to evaluate different setups trained on different datasets and approaches. We believe the adaptivity constraint still provides valuable structure to the representations, for a couple of reasons.
>
> We can compare directly with Kanzi, which is a similar diffusion decoder but without adaptivity. Generative models trained with our parameterization outperform a generative model trained with Kanzi by >25% in designability, even the models that use more sophisticated inference techniques. This observation provides a direct comparison controlling for the diffusion model; it suggests to us that the adaptivity constraint is responsible for a nontrivial amount of the performance. Training autoregressive models on Kanzi requires $L$ tokens for a lengths $L$ protein and has the same outstanding issue that other autoregressive models face; more tokens means more opportunities to missample a token, and every token takes the model further out of distribution. As we substantially outperform Kanzi, we're reasonably confident that the structure we impose is useful.
>
> It is definitely true that the diffusion decoder is a significant design choice that we made; this is part of our thesis that low resolution details can be determined by AR sampling and high resolution details should be determined by diffusion sampling. This hypothesis is supported by our new inference results, which we’ve added to the appendix – our diffusion decoder can decode accurate samples in significantly fewer steps (50 vs 250-400) than a raw diffusion decoder without conditioning. It's absolutely true that diffusion decoders generally outperform VQVAEs, at the cost of more compute. Another motivation for our approach is it obviates certain arbitrary design decisions in approaches like Kanzi (in particular, we do not need to use sliding window attention, nor is the window size a relevant hyperparameter).
>
> **Q2: Questions on scalability**
> Would you be able to clarify your comment on scalability? One point worth highlighting is we exclusively use regular transformers for our approach without SE(3) biases. Similar to diffusion models like Proteina, we believe this allows for a much more scalable architecture. Although we do not ourselves scale the model much further, for both computational and comparative reasons (scaling further would make it difficult to compare directly against Kanzi), we believe our use of simple, non-invariant architectures would facilitate scaling the model to e.g., the entire AFESM or OpenFold distillation datasets.

---

> > ### Author Rebuttal · Reviewer_L6AA · 2026-04-01
> >
> > Thank you for your reply. The question of scalability is that, compared with previous protein autoencoders like Proteina and Kanzi, which can scale up to hundreds of millions of parameters, APT seems to scale poorly beyond 47M. What is the cause of this difference?

---

> > > ### Author Response · Authors · 2026-04-03
> > >
> > > Thanks for clarifying. To be clear, **we believe APT tokens can scale**. The primary blocker to scaling protein tokenizers has historically been the use of bespoke architectures (e.g., SE(3) invariant point attention) that (1) are memory intensive and (2) are challenging to optimize and lack well established norms.
> > >
> > > The main ablation that implies otherwise is the one tagged "Stronger encoders have ambiguous performance." We've changed the language to more accurately represent our position. The main takeaway from this ablation is that the encoder, decoder, and codebook size **all need to be scaled carefully in tandem**. For example, it does no good to create a much richer codebook or more powerful encoder if the limiting factor in the model is the diffusion decoder. Importantly, this is a generic problem that applies to _all_ autoencoders, including Kanzi (which does not explore scaling to the 100M+ regime).
> > >
> > > One reason why we do not explore scaling further beyond this ablation is that a central tenet of our work is that optimizing for reconstruction is not the correct thing to do. This is something empirically observed in the literature (e.g., the structure tokenizer in DPLM2 is not very performant, but DPLM2 is one of the best masked diffusion models [1]). This is still an active area of research (e.g., it may be because of mis-samplings, or have some other cause). Nonetheless, we observe that models trained on tokens like those from Kanzi or similar local tokenizer models can quickly create unrealistic structures, even for very high fidelity tokenizers. Scaling the tokenizer would absolutely improve reconstruction performance, but potentially at the cost of downstream performance. We think the advantage of scalable architectures is (1) they're still easier to train at small scale, e.g., since APT/Kanzi/small Proteina models require much less compute and data than DPLM2/ESM3, and (2) as the field moves towards end to end training where the tokenizer can be optimized against the downstream task, scalability will matter much more.
> > >
> > > We hope this helps answer your remaining questions. If you find that the issues you raised were satisfactorily resolved, we would be grateful if you would consider updating your score.
> > >
> > > [1] Hsieh, Cheng-Yen, et al. "Elucidating the design space of multimodal protein language models." arXiv preprint arXiv:2504.11454 (2025).

---

### Official Review · Reviewer_VDgP · 2026-03-13

**Soundness:** 3
**Presentation:** 1
**Significance:** 2
**Originality:** 3
**Overall Recommendation:** 5
**Confidence:** 4

**Summary:**

This paper presents Adaptive Protein Tokenization (APT), which generates Matryoshka token embeddings for protein structures. APT is trained using an encoder-quantization-decoder pipeline with a flow-matching objective. Tail drop-out in the quantization step encourages the encoder to frontload important information. Tokens generated by APT can be applied to reconstruction, unconditional generation, and representation tasks and are competitive, but not outright superior, to standard tokenization. However, APT unlocks new possibilities such as protein compression and test-time scaling which were not possible/were not explored using existing tokenizers.

**Compliance With Llm Reviewing Policy:**

Affirmed.

**Final Justification:**

I think the technical contributions of the work are significant and enable new workflows. The rebuttal has addressed my main concerns (unfair evaluation, lack of evaluation scope, and writing quality, which the authors promise to improve), and I have raised my score.

**Key Questions For Authors:**

My takeaway is that APT might enable new workflows, but would not yield significant performance gains if integrated into existing frontier models like ESM3. Is this correct?

**Limitations:**

The authors have a limitations section.

**Strengths And Weaknesses:**

Strengths:

1. APT has a Matryoshka structure where late sequence tokens can be dropped, allowing a downstream decoder/generator to show greater creativity while still preserving crucial information.

2. The experiments are thorough and cover reconstruction, auto-regressive unconditional generation, and representation-based prediction tasks, proving that APT doesn't sacrifice performance on any front and that the Matryoshka structure applies to most, if not all, tasks.


Weaknesses:
1. The CATH-T comparison (Figure 5) is not fair. ESM-3 and DPLM representations were mean-pooled, while APT was trained to frontload information in the first token. I suggest the authors explore Pooling by Multihead Attention [1] for ESM3 and DPLM instead.

2. The authors only benchmark unconditional protein generation, which is not a biologically meaningful task. Coupled with the aforementioned CATH-T limitation, the paper's results don't adequately demonstrate APT's practical usefulness.

3.  The writing is very disorganized. Factors such as missing citations ("prior work" in L136, "prior work" in L155, etc), referencing page 7 (table 4) content on page 5, and a general lack of polish made it very hard to parse.




[1] Set Transformer: A Framework for Attention-based Permutation-Invariant Neural Networks. Juho Lee, Yoonho Lee, Jungtaek Kim, Adam R. Kosiorek, Seungjin Choi, Yee Whye Teh (2019, ICML)

---

> ### Author Rebuttal · Authors · 2026-03-30
>
> Thank you for your valuable review of our work! We are grateful for the affirmation of our experiments and results, and for your suggestions on improving our benchmarks. We discuss the latter below.
>
> **Q1: Representation learning**
> We agree that the mean-pooling is challenging for models like ESM3 and DPLM2. Our intent was not to provide an unfair comparison, but rather to emphasize this is a strength of our tokenization approach. Tasks that focus on global properties (like CATH classification, but also potentially tasks like enzyme classification and stability) benefit from this sort of hierarchical structure. Mean pooling has become the de-facto method for extracting per-structure embeddings (see [1-3]); the adaptive approach provides a convenient way around this.
>
> We performed an additional ablation where we followed the attention-pooling approach in [4]. This is equivalent to a single layer of set attention (the approach in the paper you cite), but with an additional nonlinearity and single set vector. The attention weighting improves most results slightly, but APT tokens still outperform ESM/DPLM tokens on global tasks. We present these results in our rebuttal to Reviewer FuwP.
>
> We want to highlight that during the rebuttal process, we discovered an issue with our ESM benchmarks that artificially reduced performance. This issue was related to how the public ESM models ingest structure tokens. We regret this error, and we sincerely thank the reviewer for suggesting new probing tasks which led to this discovery. We have carefully checked and updated the evaluations to reflect the new performance [6]. APT tokens still perform better than either ESM or DPLM tokens on the CATH classification task and other global tasks; the issue primarily accounts for the discrepancy between DPLM2 and ESM3.
>
> **Q2: Beyond unconditional protein generation**
>  We’ve added two additional benchmarks.
>
> 1. **Conditional generation**. We demonstrate fold-class conditioned generation (we include results for A-level conditioning, as it provides a nice balance between coarse and fine categories). We’ve added visual samples to the appendix and numerical quantifications using the fold classifier introduced in [5].
> 2. **Inference time savings**. Pure diffusion models like Proteina require >100 timesteps. AR sampling of discrete tokens is very fast and helps make maximum use of early timesteps, meaning we can use only 30-50 timesteps without losing designability. This is largely a product of our tokenizer design. The diffusion model excels at high frequency details but spends a lot of time figuring out global, low frequency structure. Tokenized AR models tend to fail because the model struggles with high frequency generation (e.g., many atoms are very slightly off, which compounds errors). Our approach allows one to interpolate between the two.
>
> **Q3: Writing improvements**
> We have done a full additional pass adding in missing citations and incorporating yours and other reviewer’s feedback for presentation. We believe these have made the paper much more coherent. We’re happy to post specific sections on the rewrite or a full changelog here if it would be helpful. In brief:
>
> 1. We've added citations throughout the paper, including to additional tokenizers and computer vision work.
> 2. We've significantly expanded background in response to Reviewer yyMG. This includes more discussion on various tokenizers, how entropy has been used in prior work/sampling, etc.
> 3. We have reorganized figures + tables to more closely match the actual flow of the paper. There are no longer any instances in the main text with tables/figures in different locations than their references.
>
> **Q4: Would APT improve performance in frontier models?**
> This is task-dependent and hard to predict. For example, models trained using APT tokens already outperform ESM3 along metrics like designability (see [5] appendix). More generally, we believe that APT tokens trained at scale would improve masked language models, since it reduces opportunities for missampling. Given that ESM3 trains on the entire AFESM and still suffers from these issues, similar techniques will likely be helpful.There are tasks like motif scaffolding for which these tokens are much worse than ESM3/DPLM2. We remark on this in our limitations section. Future frontier models will benefit from both global and local hierarchies.
>
>
> [1] https://github.com/evolutionaryscale/esm/issues/176
>
> [2] https://github.com/facebookresearch/esm See section on per-sequence embeddings
>
> [3] https://academic.oup.com/bioinformaticsadvances/article/5/1/vbaf060/8088230
>
> [4] Rao, Roshan, et al. "Evaluating protein transfer learning with TAPE."
>
> [5] Geffner, Tomas, et al. "Proteina: Scaling flow-based protein structure generative models."
>
> [6]
> | Task | ESM (MLP) | ESM (Attention) | DPLM (MLP) | DPLM (Attention) | APT (MLP) |
> |---|---:|---:|---:|---:|---:|
> | CATH-T (Acc) | 0.33 | 0.31 | 0.25 | 0.26 | 0.47 |

---

> > ### Author Rebuttal · Reviewer_VDgP · 2026-04-02
> >
> > My concerns have been fully resolved.

---

### Official Review · Reviewer_FuwP · 2026-03-21

**Soundness:** 3
**Presentation:** 3
**Significance:** 3
**Originality:** 3
**Overall Recommendation:** 5
**Confidence:** 4

**Summary:**

This paper proposes Adaptive Protein Tokenization (APT), a protein structure tokenizer that replaces local neighborhood-based tokens with a coarse-to-fine sequence of global tokens. The tokenizer is implemented as a diffusion autoencoder. Specifically, the encoder directly processes the coordinates of Ca atoms, using FSQ to produce the discrete structure tokens. A diffusion-based decoder conditions on the discrete structure tokens for reconstruction. After training an autoencoder, the obtained structure tokens are served as training target for an autoregressive language model. Comprehensive experiments evaluate APT on reconstruction, unconditional generation, representation learning, and several proof-of-concept applications such as protein shrinking and reward-guided generation. This demonstrates the effectiveness of the proposed method for comparable reconstruction and strong generation capabilities, compared with other strong baselines.

**Compliance With Llm Reviewing Policy:**

Affirmed.

**Final Justification:**

After considering both the paper and the rebuttal, I support accepting this paper. The core idea of adaptive coarse-to-fine global protein tokens is original, well motivated, and supported by strong generation results. The authors’ rebuttal addressed these concerns well by clarifying key implementation details, adding new downstream results, and providing a clearer explanation of what early versus later tokens encode structurally. Although some limitations remain, such as application for highly localized design tasks, the paper is technically solid, meaningful, and likely to inspire follow-up work in protein generative modeling.

**Key Questions For Authors:**

1. Can APT be used on infilling-style design tasks, such as motif scaffolding?

2. Relatedly, because APT uses global coarse-to-fine tokens rather than local region-specific tokens, does this make localized design more difficult? In particular, can the method support editing or redesign of a user-specified local region of a protein while preserving the rest of the structure?

**Limitations:**

Yes

**Strengths And Weaknesses:**

## Strength

1. The idea of global and adaptive structure tokenizer is novel and well-motivated. Replacing local structural tokens with ordered global tokens is an appealing direction.
2. The generation results are strong. The reported designability and scRMSD results are substantially better than other strong baselines, and the paper presents a coherent inference recipe combining token truncation, entropy-based stopping, and classifier annealing.
3. The method is practically interesting beyond reconstruction. The hierarchical prefix property leads to useful capabilities such as controllable compression, fixed-size global representations, and prefix-based inference-time search.
4. The generation of structure tokens and the reconstruction of structural coordinates are decoupled, which is an interesting paradigm to protein design. This opens up the possibility of generating a broad spectrum of proteins that meet the predefined criteria and have varying lengths, enhancing the generation diversity.

## Weakness

1. Lacks several necessary methodological details:
	- How are the discrete structural tokens injected into the diffusion/DiT decoder?
	- For the classifier annealing, how the unconditional branch, i.e. the term of v(x_t, t | $\emptyset$), is trained or obtained? Is it trained via condition dropout or a special null condition token?
	- For the autoregressive language model, is it trained on full token sequences or on truncated/dropout prefixes?
	- For the representation learning of DPLM2/ESM3, considering that they are multimodal pLMs, so what are their inputs? Only discrete structure token or both sequence and structure?

2. The author claims that early tokens store more global / low-frequency information, while later tokens store higher-frequency detail. However, the paper does not sufficiently analyze what “low-frequency” and “high-frequency” mean in the context of protein structure. If the author provides a comprehensive and interpretable analysis of what the hierarchical global tokens actually encode, it will enable readers to gain a deeper understanding of the meaning of APT tokens.
3. The representation learning evaluation is too narrow. The paper only evaluates representation learning on a single benchmark, CATH classification. The author should assess representation quality across a wider range of downstream tasks, for example, a broader collection of benchmarks provided in SaProt.

---

> ### Author Rebuttal · Authors · 2026-03-30
>
> Thank you for your thoughtful discussion and acknowledgments of the strengths of our contribution! We sincerely appreciate your feedback and we’ve done our best to address your comments below:
>
> **Q1: Additional methodological details**
> We inject the structural tokens by concatenating them with the noised sequence, then backpropagating only through the coordinate indices in the concatenated sequence. In code:
>
> ```python
> x_t = x_1 * t + (1-t) * x_0
> z_t = Linear(x_t)
> latents = cat([z_t, cfg_mask(nested_dropout(discrete_latents))])
> out = decoder(latents)
> v = out[:len(z_t)]
> ```
>
> The unconditional branch is trained by masking out the conditioning sequence with probability $p=0.1$. The autoregressive language model is trained on full token sequences. We drop tokens during inference using our entropy-based heuristics. We observed minimal difference between training on truncated vs untruncated sequences. For the representation learning, we only use the structural tokenizers for both models to conduct the probing experiments. Actual multimodal pLM trunks can do much better (and training a multimodal pLM using APT tokens is of significant interest). We've added all methodological details to the Appendix, which we hope resolves concerns
>
> **Q2. Low vs high frequency**
> Thank you for flagging this. Our goal was to appeal to intuitions from signal processing. "Low-frequency" information involves features that vary slowly over sequence position (e.g., globularity). "High frequency" information varies quickly with respect to sequence position. Readers from other fields felt that anchoring to frequency was a helpful pattern to understand what the nested dropout was doing. We have changed most of the language to "coarse" and "fine" features to avoid confusion.
>
> **Q3: Additional representation learning tasks**
> We have added the following benchmarks (see [3] for numerical comparisons). On global tasks like EC prediction or CATH [1], APT tokens outperform ESM/DPLM, but on local tasks, there is much more variance.
>
> 1. **Enzyme commission classification**. APT tokens achieve F1 scores > 0.30 (compare to ESM3/DPLM2, which have F1 scores of approximately 0.10). This includes additional probing based on attention, which Reviewer VDgP requested.
> 2. **GO classification** There is much higher variance, as many GO annotations depend more on sequence than structure. APT tokens are competitive.
> 3. APT tokens allow one to use simple **vector embeddings instead of alignments** for structural clustering, which is much faster and enables new applications (for instance, backpropagating through kNN searches, retrieval augmented generation, etc.). To measure this, we clustered proteins using Foldseek and constructed clusters using k-means over our APT embeddings. The average within-cluster TMscore for Foldseek was 0.5136, and the within-cluster TMscore for APT clusters centered between 0.48 and 0.53 depending on the number of clusters (see [2]).
> 4. **Metal ion binding** classification. This is a local property prediction task (are there metal ion binding sites) [2], where underperforms local tokenizers (as expected, se [3]).
> 5. In response to VDgP, we have added **attention probing** to our ESM/DPLM results. These  improve performance slightly but do not significantly change results.
>
> **Q4: Performance on local tasks**
> Correct, APT tokens are not useful for tasks like motif scaffolding. This is a great point; we mention this  in the limitations section. The current paradigm of locality is well suited for some but not all tasks. Future frontier models would benefit by incorporating multiple hierarchies of information. To emphasize this point, we added a metal ion binding task.
>
>
> [1] Rao, Roshan, et al. "Evaluating protein transfer learning with TAPE."
> [2] Foldseek retrieval experiment. We generate clusters at different $k-means$ with APT tokens and consider two quantities. The weighted TM mean within a cluster represents the average similarity, and the fraction of pairs with TMscore > 0.50 reflects clusters with the same fold. We do not advocate to use APT tokens as a direct replacement for Foldseek; the Foldseek VQVAE is trained as to preserve evolutionary descriptors, not just consider structure. These results indicate that training Foldseek style features in a manner similar to APT tokens would provide the best of both worlds.
>
> | Method | Kept clusters (>=8) | Weighted TM mean | Weighted frac scored pairs TM > 0.50 |
> |---|---:|---:|---:|
> | Foldseek | 217 | 0.514 | 0.506 |
> | APT k=32 | 32 | 0.486 | 0.39 |
> | APT k=64 | 64 | 0.5 | 0.415 |
> | APT k=128 | 128 | 0.513 | 0.454 |
> | APT k=256 | 256 | 0.54 | 0.518 |
>
> [3] **Numerical results on new representation tasks**
>
> | Model | EC (Fmax) | Metal ion binding (Acc) | GO (Fmax) |
> |---|---:|---:|---:|
> | APT (MLP) | 0.31 | 0.52 | 0.29 |
> | ESM (MLP) | 0.085 | 0.56 | 0.28 |
> | ESM (Attention) | 0.073 | 0.56 | 0.28 |
> | DPLM (MLP) | 0.13 | 0.62 | 0.26 |
> | DPLM (Attention) | 0.08 | 0.57 | 0.27 |

---

> > ### Author Rebuttal · Reviewer_FuwP · 2026-04-03
> >
> > Thank you for the detailed rebuttal and the additional experiments. I have read the response carefully and appreciate the clarifications provided.
> >
> > I still have two questions that are not entirely clear to me.
> >
> > First, I am very interested in the specific meaning encoded by the structural tokens. Could the authors provide a more detailed analysis of what these structural tokens actually represent? In particular, it would be helpful to include some form of visualization or interpretability analysis showing what different structural tokens, especially coarse-grained versus fine-grained ones, correspond to in real protein structures. For example, do some coarse tokens capture overall shape or fold-level information, while finer tokens capture local geometric details? A more interpretable analysis along these lines would significantly improve my understanding of the method. If the authors can provide such evidence, I would consider increasing my score.
> >
> > Second, could the authors discuss possible ways to extend APT to infilling tasks? Infilling is a common generation scenario, and it would be helpful to understand whether the current framework could be adapted to such a scenario, or what modifications might be needed to make this possible.

---

> > > ### Author Response · Authors · 2026-04-04
> > >
> > > **The meaning of structural tokens:** This is a great question; Figure 10 in the Appendix H is a helpful point of reference. If you look at that figure, we've plotted a few visualizations of proteins across different token strengths. There are a few observations we highlight.
> > >
> > > First, at 64 tokens, 99.4% of proteins have TMscore greater than 0.5, which is a useful heuristic for capturing the same fold. Smaller adjustments, which heavily contribute to better RMSD scores, emerge as we add a greater number of tokens. Visually, there's very little difference past 64 tokens because most of the information goes into making small, local adjustments (you can see this in particular in the coils). So we think your intuition that early tokens contribute heavily to the overall fold is accurate.
> > >
> > > The other way to view the tokens is via secondary structure. One thing we've thought about a lot in this work is the intrinsic complexity of a protein. A problem we wanted to address was that the current tokenization paradigm uses many tokens to encode large simple proteins and few tokens to encode small, complicated proteins. This maps on well to secondary structure. **Highly flexible coils have much more information than a large but relatively rigid alpha helix**; every alpha helix is basically the same size and overall structure, with perhaps some low frequency undulation (meaning the structure changes slowly over the length of the protein, look at PDB 1C1G for a good example). To see an example, look at the fourth row in Figure 10. As the model gets more tokens, it can more accurately represent the complicated, unpredictable coils, which it initially interprets as alpha helices. **Later fine-grained tokens provide more information about highly disordered segments that are primarily coils, and earlier coarse-grained tokens provide more information about very structured regions like alpha helices**.
> > >
> > > We also want to highlight that global structure, set by earlier tokens, can affect downstream local structure. As a concrete example, the protein in the second row is mostly beta sheets. The 32 token example encodes most of the global structure correctly, so most of the beta sheets are reconstructed because that regular repetitive global pattern is very unique to beta sheets (so the diffusion decoder has a very easy time). However, those 32 tokens aren't enough in this case, the tokenizer gets one segment misaligned. The diffusion decoder decodes this to an alpha helix, because it has global structure that's more consistent with an alpha helix. As we add more tokens, the secondary structure gets filled in properly.
> > >
> > > We can quantify this further -- for instance, we can make plots showing the percentage of coil increases as the number of tokens increases. This partially motivates the low/high frequency terminology; the random "squiggles" of coils tend to be described by later tokens, and they initially appear as simple, straight line alpha helices.
> > >
> > > **Extending to infilling:** Infilling is actually pretty easy -- for a small missing segment, the overall shape of the protein doesn't change much, so one can just encode the structure with a straight line approximation for the missing piece, then throw away the tail to remove the effect of the approximation and decode the structure with some classifier-free annealing. The diffusion decoder will fill in a plausible local pattern. We can demonstrate this on some small examples if you view this as important.
> > >
> > > Despite what we said earlier, it's actually possible to do motif scaffolding, it's just a bit harder. Two ways we can do this
> > > 1. Joint local + global tokens. We think this is definitely an option, local tokens will probably benefit from having global conditioning. This is a bit inelegant, though, and has other problems [1]
> > > 2. Train a classifier that detects the presence of the motif, and use this coupled with the beam search to generate. The upside of this approach is the model learns where to place motifs. The downside is training the classifier is probably harder than just training a standard diffusion motif scaffolding model.
> > >
> > > We hope this is helpful! We have tried to answer as thoroughly as possible due to the limit on responses to reviewers; if this is satisfactory, we hope you will consider raising your score. Thank you!
> > >
> > > [1] Current tokenized models generally do motif scaffolding by removing tokens _after_ tokenizing the model, which means some information about the surroundings has usually leaked into the initial motif tokens.

---

### Decision · Program_Chairs · 2026-04-30

**Decision:**

Accept (regular)

**Comment:**

The reviewer discussion was broadly positive about the central contribution. Reviewers found the idea of replacing local structural tokens with a coarse-to-fine sequence of global adaptive tokens to be novel and technically meaningful. The strongest support came from the reconstruction and generation results. Several reviewers also viewed the representation itself as useful beyond those tasks, particularly because it allows controllable compression and fixed-size global embeddings.

The main concerns were: (1) presentation and accessibility, including missing methodological detail; (2) the scope and fairness of the empirical evaluation, especially for representation learning; and (3) the boundaries of the claimed contribution, in particular how much of the gain should be attributed to the tokenizer rather than the diffusion decoder and how well the method applies to localized design tasks.

The rebuttal addressed most of these issues in a substantive way. It clarified the training and conditioning setup, added further representation and generation results, responded to the fairness concerns around pooling, and made the intended scope of the method clearer. Some reservations remained, especially around accessibility and framing, but the post-rebuttal discussion nevertheless moved in a more favorable direction.

The paper is not without weaknesses, and the final version would benefit from improved clarity. That said, the core contribution is sound, non-redundant, and relevant to a meaningful part of the ICML community. I have also taken the authors' rebuttal and the subsequent discussion into account in reaching this decision. Overall, I recommend acceptance.